# SDformer: Similarity-driven Discrete Transformer For Time Series Generation

**Zhicheng Chen**[1,2,†]**, Shibo Feng**[3]**, Zhong Zhang**[2]**, Xi Xiao**[1,4,*]**, Xingyu Gao**[5]**, Peilin Zhao**[2,*]

[1]Shenzhen International Graduate School, Tsinghua University
[2]Tencent AI Lab
[3]School of Computer Science and Engineering, Nanyang Technological University
[4]Key Laboratory of Data Protection and Intelligent Management (Sichuan University), Ministry of Education
[5]Institute of Microelectronics, Chinese Academy of Sciences
{czc22@mails,xiaox@sz}.tsinghua.edu.cn, shibo001@ntu.edu.sg,
gaoxingyu@ime.ac.cn, {todzhang, masonzhao}@tencent.com

## Abstract

The superior generation capabilities of Denoised Diffusion Probabilistic Models (DDPMs) have been effectively showcased across a multitude of domains. Recently, the application of DDPMs has extended to time series generation tasks, where they have significantly outperformed other deep generative models, often by a substantial margin. However, we have discovered two main challenges with these methods: 1) the inference time is excessively long; 2) there is potential for improvement in the quality of the generated time series. In this paper, we propose a method based on discrete token modeling technique called Similarity-driven Discrete Transformer (SDformer). Specifically, SDformer utilizes a similarity-driven vector quantization method for learning high-quality discrete token representations of time series, followed by a discrete Transformer for data distribution modeling at the token level. Comprehensive experiments show that our method significantly outperforms competing approaches in terms of the generated time series quality while also ensuring a short inference time. Furthermore, without requiring retraining, SDformer can be directly applied to predictive tasks and still achieve commendable results.

## 1 Introduction

Time series data is prevalent across a wide array of real-world applications, spanning fields such as finance [22, 9, 38, 36], healthcare [27], energy [28, 21, 12, 13], retail [20, 45], and climate science [35] . Despite its significance, the limited availability of dynamic data can pose a significant barrier to the development of machine learning solutions, particularly in scenarios where data sharing could lead to privacy violations [2]. The generation of synthetic yet realistic time series data has emerged as a promising alternative, garnering increased interest due to recent advancements in deep learning techniques.

Existing works on time series generation (TSG) is mainly based on common deep generative models, such as methods based on generative adversarial networks (GAN) [24, 11, 39, 37, 26, 16, 17] and methods based on Variational Autoencoders (VAE) [7, 25]. Currently holding state-of-the-art results are DDPMs-based methods [19, 6, 42], which have the capability to generate high-quality, realistic time series. However, they are not without their challenges. Firstly, these methods often require lengthy inference times due to the substantial number of denoising steps involved. Secondly, despite

---

[†]This work is done when Zhicheng Chen works as an intern in Tencent AI Lab.
[*]Corresponding author.

38th Conference on Neural Information Processing Systems (NeurIPS 2024).

these methods achieving significant advancements in generation quality compared to other deep generative models, we observe that the quality of the time series they generate still has potential for further enhancement.

Currently, large transformer-based language models, often known as LMs or LLMs, have become the standard choice for natural language generation tasks[1, 3]. As time has progressed, these LMs have evolved to produce content across a wide range of modalities, such as images [29, 40, 5, 4] and videos [41, 33], using what is referred to as Discrete Token Modeling (DTM) technique. In general, these approaches function by learning a discrete representations of images (or videos, etc.), treating them as if they were natural language, and harnessing the power of existing language models for the generation process. Inspired by their success, we aim to explore the application of these techniques in the domain of multivariate time series generation, potentially unlocking new possibilities and advancements in this area.

To address the aforementioned challenges, we propose a novel two-stage method for time series generation, called Similarity-driven Discrete Transformer (SDformer). The primary objective of the first stage is to employ Vector Quantized Variational Autoencoders (VQ-VAE) [31] for learning high-quality discrete representations of time series. To enhance this process, we introduce a similarity-driven vector quantization approach, which identifies the most suitable code from the codebook by maximizing similarity. The experiments in 5.4 further substantiate the superiority of our method over distance-driven vector quantization. Moreover, to prevent code collapse—a phenomenon where only a small portion of the codes are updated during training, thereby hindering the performance of the VQ-VAE—we incorporate two standard recipes [34] during training: Exponential Moving Average (EMA) for codebook updates and Resetting inactivated codes during the training process (Code Reset). In the second stage, we implement two Discrete Token Modeling (DTM) techniques: Masked Token Modeling (MTM) and Autoregressive Token Modeling (ARTM), underpinning the SDformer-ar and SDformer-m variants, respectively. SDformer-ar adopts an autoregressive approach for both training and inference, mitigating the inconsistency between these two phases through random replacement [43]. SDformer-m utilizes random masking for training and iterative decoding for inference [8, 5]. Our findings reveal that SDformer, particularly SDformer-ar, surpasses existing models in time series generation. Moreover, SDformer demonstrates robust predictive performance without requiring retraining.

In summary, our contributions include:

- We propose an efficient time series generation model **SDformer**, which successfully introduces DTM technique into time series generation and demonstrates its feasibility and efficiency.

- We introduce a novel similarity-driven vector quantization approach that outperforms the traditional distance-driven method in learning discrete representations of time series. This innovative approach offers a straightforward yet powerful technique for applying discrete token modeling in various fields.

- Our experimental results confirm that the SDformer's performance in time series generation notably surpasses that of the current state-of-the-art models, exemplified by an average enhancement of 60.8% in Discriminative Score and 86.5% in Context-FID Score across multiple datasets.

## 2 Related work

### 2.1 Discrete token modeling

Discrete token modeling, a staple in natural language processing (NLP), has recently been adapted for non-language modalities through vector quantization models like VQ-VAE [31] and VQGAN [10]. These models enable the encoding of diverse data types into discrete tokens, allowing the application of advanced language modeling techniques to generate content across various domains. This expansion significantly broadens the utility of NLP methodologies, extending their impact beyond traditional language tasks. Among these techniques, Autoregressive Token Modeling (ARTM) is a common approach that predicts the next token in a sequence, given the previous tokens, using a categorical distribution. Models such as DALL-E [29] and Parti [40] employ ARTM to accomplish text-to-image generation tasks. Similarly, T2M-GPT [43] and MotionGPT [18] utilize ARTM for

text-to-motion generation. Another widely used technique is Masked Token Modeling (MTM), which is trained using a masked token objective [8]. In this approach, some tokens in the sequence are randomly masked and need to be predicted based on the observed tokens. Models such as MaskGIT [5] and MUSE [4] leverage MTM for image generation tasks. Furthermore, MAGVIT [41] and Phenaki [33] employ MTM for video generation, showcasing the versatility of this technique.

## 2.2 Time series generation

Deep generative models demonstrate high-quality sample generation across various domains, as does time series generation. At first, people mostly relied on GAN to complete time series generation [24, 11, 39, 37, 26, 16]. For example, TimeGAN [39] improves temporal dynamics capture by adding an embedding function and supervised loss. COT-GAN [37] combines GAN and Causal Optimal Transfer (COT) principles to efficiently and stably generate low- and high-dimensional time series data. Due to the challenges of training instability and mode collapse in GAN, researchers have started exploring alternative deep generative models for TSG. TimeVAE [7] employs an interpretable temporal structure and achieves promising results in time series synthesis using VAE. Moreover, several studies focus on addressing the generation of irregular time series, such as GT-GAN [17] and KoVAE [25].

With the emergence of Denoising Diffusion Probabilistic Models (DDPMs) [14], a new class of generative models, impressive generative capabilities have been demonstrated across various domains. Recently, diffusion models have also been adapted for TSG. For instance, DiffWave [19] directly applies DDPMs to waveform generation, while DiffTime [6] harnesses the latest advancements in score-based diffusion models for time series generation. Furthermore, Diffusion-TS [42] generates time series samples by utilizing an encoder-decoder transformer with disentangled temporal representations, showcasing the versatility and potential of these alternative generative models.

## 3 Definitions and problem formulation

We define multivariate time series as $X_{1:\tau} = (x_1, \cdots, x_\tau) \in \mathbb{R}^{\tau \times d}$, where $\tau$ and $d$ are the number of time steps and variables respectively. Assuming that a dataset containing $n$ time series can be expressed as $D = \{X_{1:\tau}^i\}_{i=1}^n$, the goal of unconditional generation is to use a model $f_\theta$ to generate time series with the same distribution as $D$, i.e.,

$$\hat{X}_{1:\tau}^i = f_\theta(Z), \tag{1}$$

where $Z$ is the input sampled from any known distribution, such as the Gaussian distribution.

Time series forecasting is a common conditional time series generation. We denote historical values as $X_{1:l} \in \mathbb{R}^{l \times d}$, where $1 < l < \tau$ is the number of historical time steps. Therefore, the goal of conditional generation is to use a model $f_\theta$ to predict future values, i.e.,

$$\hat{X}_{l+1:\tau} = f_\theta(X_{1:l}). \tag{2}$$

In this paper, our objective is to develop an effective approach that not only accomplishes unconditional generation tasks efficiently but also adapts to conditional generation tasks without retraining, while maintaining high accuracy.

## 4 Methods

In this section, we illustrate proposed innovative model SDformer for time series generation. Specifically, SDformer is a two-stage method, the framework of which is illustrated in Figure 1. In the first stage, a pre-trained time series tokenizer utilizes similarity-driven vector quantization to obtain high-quality discrete token representations. Following this, a discrete Transformer is employed to learn the distribution of time series data at the discrete token level, with the two generative ways (Masked and Autoregressive strategies).

### 4.1 Time series tokenizer

To represent time series in discrete tokens, we pre-train a multivariate time series tokenizer based on the VQ-VAE architecture [31]. Our time series tokenizer consists of an encoder $\mathcal{E}$ and a decoder

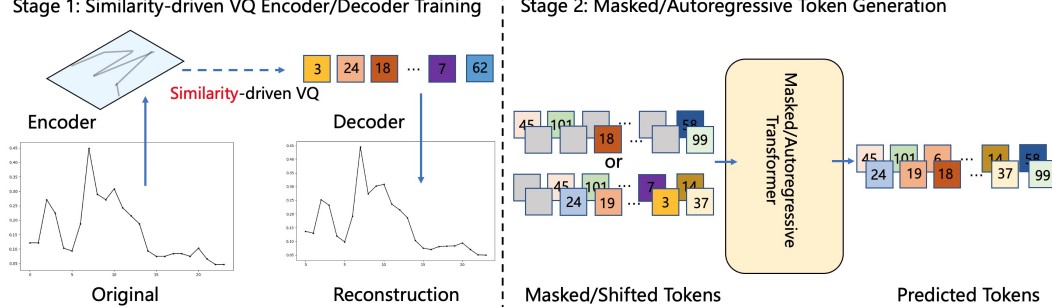

Figure 1: The workflow of SDformer. In stage 1, we pre-train a time series tokenizer which uses similarity-driven vector quantization to obtain high-quality discrete token representations. In stage2, two optional techniques are introduced for time series modeling at the discrete token level: MTM and ARTM. For MTM, the input tokens are randomly masked and fed into the Masked Transformer, an encoder-only model, to predict the masked tokens. Conversely, for ARTM, the input tokens are shifted back by one step with the [BOS] token added at the starting position, and then processed by the Autoregressive Transformer, a decoder-only model, to predict subsequent tokens for all input tokens.

$\mathcal{D}$. The encoder is responsible for the generation of discrete time series tokens, while the decoder is capable of reconstructing these tokens back into their original time series form. This methodology allows us to represent time series akin to a language, thereby enabling the application of a multitude of efficient language models to address various time series-related tasks.

Specifically, the encoder $\mathcal{E}$ initially applies 1D convolutions to time series features $X_{1:\tau}$ along the temporal dimension, resulting in latent vectors $H_{1:L} = (h_1, \cdots, h_L) \in \mathbb{R}^{L \times d_c}$, where $L = \tau/r$, $r$ signifies the temporal downsampling rate and $d_c$ is hidden dimension. Subsequently, we employ the codebook to discretely quantize $h_i$ to obtain discrete token. The learnable codebook $C = \{c_k\}_{k=0}^{K-1} \subset \mathbb{R}^{d_c}$ comprises $K$ latent embedding vectors, each with a dimension $d_c$. The process of similarity-driven vector quantization $Q(\cdot)$ involves identifying the index of the vector in the codebook that exhibits the highest similarity to $h_i$, which can be expressed as:

$$y_i = Q(h_i) := \underset{k=0,\cdots,K-1}{\arg\max} \frac{h_i}{||h_i||} \cdot \frac{c_k}{||c_k||}, \tag{3}$$

where $y_i = 0, \cdots, K-1$, $\cdot$ denotes the inner product, and $||\cdot||$ represents the modulo operation. For simplicity, we introduce a normalization step in the final output layer of the encoder $\mathcal{E}$, resulting in a unit modulus length for $h_i$. Furthermore, we ensure that the code in the codebook always has a unit modulus length for $c_k$. The similarity-driven quantization process can be re-simplified as:

$$y_i = \underset{k=0,\cdots,K-1}{\arg\max} h_i \cdot c_k. \tag{4}$$

Following quantization, the dequantization process $Q^{-1}(\cdot)$ reverts $y_i$ back to the latent embedding vector, denoted as:

$$\tilde{h}_i = Q^{-1}(y_i) := c_{y_i}. \tag{5}$$

Ultimately, the decoder $\mathcal{D}$ restores it to the raw time series space, i.e., $\tilde{X}_{1:\tau} = \mathcal{D}(\tilde{H}_{1:L})$. To train this time series tokenizer, we utilize two distinct loss functions for training and optimizing the parameters of $\mathcal{E}$ and $\mathcal{D}$:

$$\mathcal{L} = ||X_{1:\tau} - \tilde{X}_{1:\tau}||_2^2 + \frac{\lambda}{L} \sum_{i=1}^{L} \left(1 - h_i \cdot sg(\tilde{h}_i)\right), \tag{6}$$

where the first loss is the reconstruction loss, the second loss is embedding loss, $sg(\cdot)$ represents the stop gradient, and $\lambda$ is hyperparameter used to adjust the weights of different parts. For the codebook, we use Exponential Moving Average and Codebook Reset techniques [34] to update.

When the time series tokenizer training is completed, the codebook and all parameters will be frozen. By employing this time series tokenizer, a multivariate time series $X_{1:\tau} \in \mathbb{R}^{\tau \times d}$ can be mapped to a

sequence of time series tokens $Y_{1:L} \in \{0, \cdots, K-1\}^L$. Therefore, we can use DTM technique to learn the distribution of time series data at the discrete token level. For the choice of DTM, we can opt for methods such as ARTM or MTM. We will introduce these two methods in Sections 4.2 and 4.3, respectively.

## 4.2 Autoregressive token modeling on time series generation

In this part, we utilize ARTM technique to learn the distribution of time series data at the discrete token level, based on the time series tokenizer. We refer to this approach as SDformer-ar. During training, we take shifted tokens $Y_{1:L}^{in} = ([BOS], y_1, \cdots, y_{L-1})$ as input and real tokens $Y_{1:L}$ as target for training, where [BOS] represents Beginning of Sentence token. In particular, we use index $K$ as the [BOS] token, which is distinct from the codebook's index range $\{0, \cdots, K-1\}$. The training objective is to minimize the negative log-likelihood of all tokens:

$$\mathcal{L}_{ar} = -\mathbb{E} \left[ \sum_i \log P(y_i | Y_{1:i}^{in}) \right]. \tag{7}$$

Concretely, we input $Y_{1:L}^{in}$ into a Decoder-only Transformer to predict the probabilities $P(y_i | Y_{1:i}^{in})$ for each token, where the negative log-likelihood is computed as the cross-entropy between the ground-truth one-hot token and predicted token. For inference, we start from the [BOS] token and generate next token in an autoregressive fashion. The detailed training and inference algorithm of SDformer-ar are respectively shown in Algorithm 2 and 4 in Appendix E. Note that we are able to generate diverse time series by sampling from the predicted distributions given by the transformer.

**Random replacement.** Autoregression is known to exhibit inconsistency between the training and inference phases. Specifically, during training, the first $i-1$ ground-truth tokens are used to predict the $i$-th token. However, during inference, there's no guarantee that all the preceding tokens used as conditions are correct. To alleviate this issue, we implement a random replacement strategy as a form of data augmentation during training. In this approach, each token is processed individually. A random number is compared to a probability threshold $\pi$. If it meets the threshold, the token is replaced randomly; otherwise, it remains unchanged. The random replacement can be expressed as:

$$\tilde{y}_i = \begin{cases} \text{Randint}(0, K), & \text{if Uniform}(0, I) \leq \pi \\ y_i, & \text{otherwise} \end{cases}, \tag{8}$$

where $\text{Randint}(0, K)$ is a random integer sampled uniformly from the range $0$ to $K-1$, and $\text{Uniform}(0, I)$ is a random number sampled from a uniform distribution in the range $0$ to $1$. Therefore, during training, we use $\tilde{y}_i$ instead of $y_i$ in $Y_{1:L}^{in}$ to achieve data augmentation.

## 4.3 Masked token modeling on time series generation

In this part, we utilize MTM technique to learn the distribution of time series data at the discrete token level, based on the time series tokenizer. We refer to this approach as SDformer-m. During training, we sample a probability $p$ from the uniform distribution $U(0, 1)$ as the mask probability. We then replace tokens in the original token sequence with the [MASK] token according to the mask probability $p$. In particular, we use index $K$ as the [MASK] token, which is distinct from the codebook's index range $\{0, \cdots, K-1\}$. In other words, when the token $y_i = K$ at a certain position, it indicates that the position has been masked. Denote $\overline{Y}_{1:L}$ as the result after applying random mask to $Y_{1:L}$. The training objective is to minimize the negative log-likelihood of the masked tokens:

$$\mathcal{L}_{mask} = -\mathbb{E} \left[ \sum_{\overline{y}_i = K} \log P(y_i | \overline{Y}_{1:L}) \right]. \tag{9}$$

Concretely, we feed the masked $\overline{Y}_{1:L}$ into a multi-layer bi-directional transformer to predict the probabilities $P(y_i | \overline{Y}_{1:L})$ for each masked token, where the negative log-likelihood is computed as the cross-entropy between the ground-truth one-hot token and predicted token.

During inference, we generate a new token sequence using iterative decoding, as proposed in [5]. Initially, we set an iteration number $N$ and a mask schedule $S$ of length $N$. Here, $S[t]$ represents the number of masks needed after the completion of step $t$. It is required that $S[0] < N$, $S[T-1] = 0$, and $S[t]$ strictly decreases with an increase in $t$. The detailed training and inference algorithm of SDformer-m are respectively shown in Algorithm 3 and 5 in Appendix E.

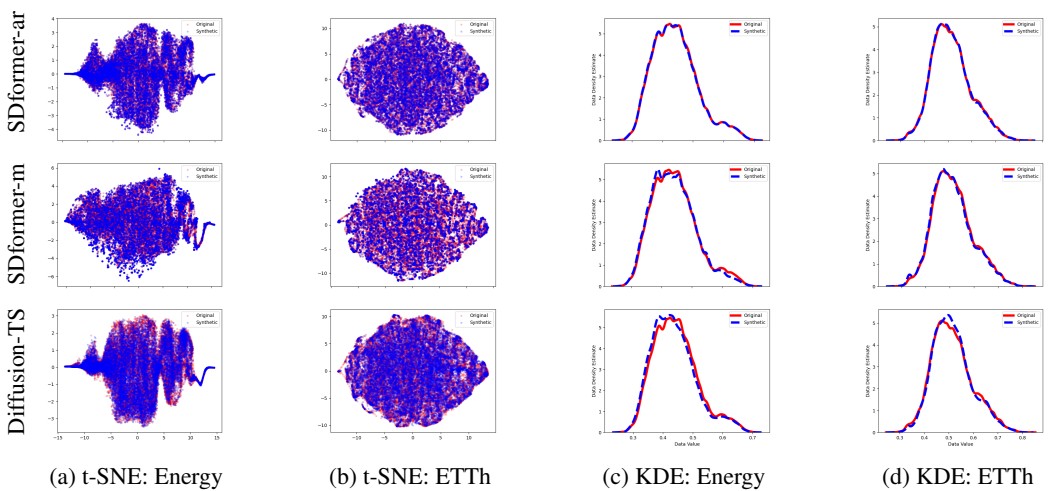

| (a) t-SNE: Energy | (b) t-SNE: ETTh | (c) KDE: Energy | (d) KDE: ETTh |

Figure 2: Visualizations of the time series synthesized by SDformer and Diffusion-TS.

Table 1: Descriptions of all datasets.

| Dataset | Sines | Stocks | ETTh | MuJoCo | Energy | fMRI |
|---|---|---|---|---|---|---|
| # of Samples | 10000 | 3773 | 17420 | 10000 | 19711 | 10000 |
| dim | 5 | 6 | 7 | 14 | 28 | 50 |

## 5 Experiments

In this section, we commence by assessing our proposed methods through a comparative analysis with several state-of-the-art baseline methods on unconditional time series generation tasks. Subsequently, we delve deeper into the analysis of our methods' versatility and high performance in conditional generation tasks. Lastly, through ablation experiments, we confirm the superior effectiveness of similarity-driven vector quantization and discrete token modeling.

### 5.1 Experimental setups

**Datasets** To evaluate the performance of SDformer, we conduct experiments on 4 real-world datasets (Stocks, ETTh, Energy and fMRI) and 2 simulated datasets (Sines and MuJoCo). Table 1 provides a partial description of each dataset. For more detailed information, please refer to Appendix A.

**Metrics** For quantitative evaluation of synthesized data, we employ the discriminative score and predictive score as described in [39], along with the Context-FID score proposed by [16]. For detailed descriptions, please refer to Appendix A.

### 5.2 Unconditional time series generation

Table 2 provides a summary of the performance for each of the compared algorithms on all the datasets. From these results, we can make several observations. Firstly, our proposed methods based on DTM outperform other methods in most cases, demonstrating the feasibility and effectiveness of DTM for time series generation tasks. Secondly, SDformer-ar exhibits a significantly better performance than SDformer-m, which contrasts with the findings in the visual domain. This can be attributed to the fact that autoregressive token modeling is better suited to capture temporal correlations compared to masked token modeling.

To further investigate the capability of our proposed methods in handling longer sequences, we compare the generative abilities of different methods on longer time series, as shown in Table 3. Based on the results, it is evident that many methods exhibit significant distortions when dealing with longer time series, particularly when the discriminative score approaches 0.5, as seen prominently

Table 2: Results of all methods on all datasets

| Metrics | Methods | Sines | Stocks | ETTh | MuJoCo | Energy | fMRI |
|---------|---------|-------|--------|------|--------|--------|------|
| Discriminative Score↓ | SDformer-ar | **0.006±.004** | **0.010±.006** | **0.003±.001** | **0.008±.005** | **0.006±.004** | **0.017±.007** |
| | SDformer-m | 0.008±.004 | 0.020±.011 | 0.022±.001 | 0.0250±.007 | 0.062±.006 | 0.043±.006 |
| | Diffusion-TS | **0.006±.007** | 0.067±.015 | 0.061±.009 | **0.008±.002** | 0.122±.003 | 0.167±.023 |
| | TimeGAN | 0.011±.008 | 0.102±.021 | 0.114±.055 | 0.238±.068 | 0.236±.012 | 0.484±.042 |
| | TimeVAE | 0.041±.044 | 0.145±.120 | 0.209±.058 | 0.230±.102 | 0.499±.000 | 0.476±.044 |
| | Diffwave | 0.017±.008 | 0.232±.061 | 0.190±.008 | 0.203±.096 | 0.493±.004 | 0.402±.029 |
| | DiffTime | 0.013±.006 | 0.097±.016 | 0.100±.007 | 0.154±.045 | 0.445±.004 | 0.245±.051 |
| | Cot-GAN | 0.254±.137 | 0.230±.016 | 0.325±.099 | 0.426±.022 | 0.498±.002 | 0.492±.018 |
| Predictive Score↓ | SDformer-ar | **0.093±.000** | 0.037±.000 | **0.118±.002** | **0.007±.001** | **0.249±.000** | **0.091±.002** |
| | SDformer-m | **0.093±.000** | 0.037±.000 | 0.119±.002 | **0.007±.001** | 0.250±.000 | **0.091±.001** |
| | Diffusion-TS | **0.093±.000** | **0.036±.000** | 0.119±.002 | **0.007±.000** | 0.250±.000 | 0.099±.000 |
| | TimeGAN | **0.093±.019** | 0.038±.001 | 0.124±.001 | 0.025±.003 | 0.273±.004 | 0.126±.002 |
| | TimeVAE | **0.093±.000** | 0.039±.000 | 0.126±.004 | 0.012±.002 | 0.292±.000 | 0.113±.003 |
| | Diffwave | **0.093±.000** | 0.047±.000 | 0.130±.001 | 0.013±.000 | 0.251±.000 | 0.101±.000 |
| | DiffTime | **0.093±.000** | 0.038±.001 | 0.121±.004 | 0.010±.001 | 0.252±.000 | 0.100±.000 |
| | Cot-GAN | 0.100±.000 | 0.047±.001 | 0.129±.000 | 0.068±.009 | 0.259±.000 | 0.185±.003 |
| | Original | 0.094±.001 | 0.036±.001 | 0.121±.005 | 0.007±.001 | 0.250±.003 | 0.090±.001 |
| Context-FID Score↓ | SDformer-ar | **0.001±.000** | **0.002±.000** | **0.008±.001** | **0.005±.001** | **0.003±.000** | **0.015±.001** |
| | SDformer-m | 0.010±.002 | 0.034±.008 | 0.019±.003 | 0.030±.003 | 0.041±.005 | 0.035±.003 |
| | Diffusion-TS | 0.006±.000 | 0.147±.025 | 0.116±.010 | 0.013±.001 | 0.089±.024 | 0.105±.006 |
| | TimeGAN | 0.101±.014 | 0.103±.013 | 0.300±.013 | 0.563±.052 | 0.767±.103 | 1.292±.218 |
| | TimeVAE | 0.307±.060 | 0.215±.035 | 0.805±.186 | 0.251±.015 | 1.631±.142 | 14.449±.969 |
| | Diffwave | 0.014±.002 | 0.232±.032 | 0.873±.061 | 0.393±.041 | 1.031±.131 | 0.244±.018 |
| | DiffTime | 0.006±.001 | 0.236±.074 | 0.299±.044 | 0.188±.028 | 0.279±.045 | 0.340±.015 |
| | Cot-GAN | 1.337±.068 | 0.408±.086 | 0.980±.071 | 1.094±.079 | 1.039±.028 | 7.813±.550 |

in the Energy dataset. Despite these challenges, both SDformer-ar and SDformer-m continue to demonstrate exceptional performance.

To visualize the performance of time series generation, we adopt two visualization methods: projecting original and synthetic data in a 2-dimensional space using t-SNE [32], and drawing data distributions using Kernel Density Estimation (KDE). Figure 2 illustrates the visualization of our methods in comparison with Diffusion-TS on the Energy and ETTh datasets, revealing that the data generated by SDformer-ar more closely resembles the real data, followed by SDformer-m.

## 5.3 Conditional time series generation

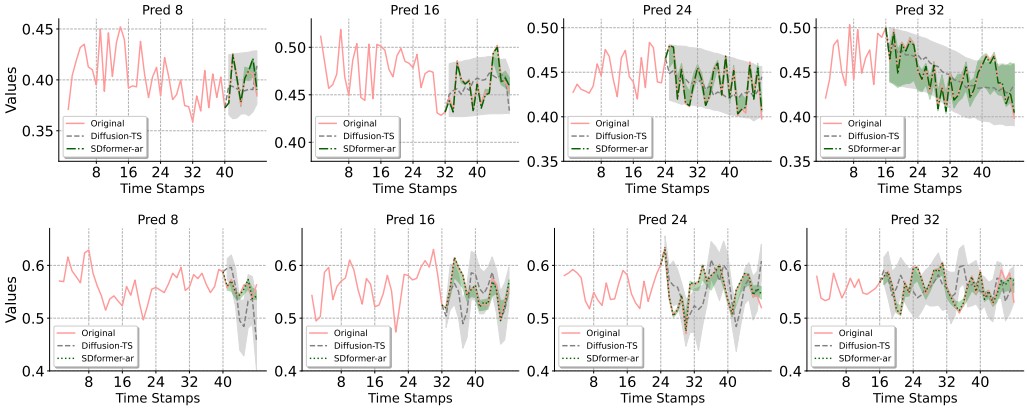

Figure 3: Examples of time series forecasting for Energy ($1^{st}$ row) and fMRI ($2^{st}$ row) datasets. Green and gray colors correspond to SDformer-ar and Diffusion-TS, respectively.

Table 3: Results of long-term time series generation

| Metrics | Methods | ETTh | | | Energy | | |
|---|---|---|---|---|---|---|---|
| | | 64 | 128 | 256 | 64 | 128 | 256 |
| Discriminative Score↓ | SDformer-ar | **0.018±.007** | **0.013±.005** | **0.008±.006** | **0.010±.007** | **0.013±.007** | **0.017±.003** |
| | SDformer-m | 0.034±.017 | 0.038±.008 | 0.041±.024 | 0.053±.018 | 0.069±.014 | 0.035±.007 |
| | Diffusion-TS | 0.106±.048 | 0.144±.060 | 0.060±.030 | 0.078±.021 | 0.143±.075 | 0.290±.123 |
| | TimeGAN | 0.227±.078 | 0.188±.074 | 0.442±.056 | 0.498±.001 | 0.499±.001 | 0.499±.000 |
| | TimeVAE | 0.171±.142 | 0.154±.087 | 0.178±.076 | 0.499±.000 | 0.499±.000 | 0.499±.000 |
| | Diffwave | 0.254±.074 | 0.274±.047 | 0.304±.068 | 0.497±.004 | 0.499±.001 | 0.499±.000 |
| | DiffTime | 0.150±.003 | 0.176±.015 | 0.243±.005 | 0.328±.031 | 0.396±.024 | 0.437±.095 |
| | Cot-GAN | 0.296±.348 | 0.451±.080 | 0.461±.010 | 0.499±.001 | 0.499±.001 | 0.498±.004 |
| Predictive Score↓ | SDformer-ar | **0.116±.006** | 0.110±.007 | **0.095±.003** | **0.247±.001** | **0.244±.000** | **0.243±.002** |
| | SDformer-m | 0.120±.004 | **0.107±.004** | 0.110±.007 | 0.248±.001 | 0.245±.000 | 0.244±.003 |
| | Diffusion-TS | **0.116±.000** | 0.110±.003 | 0.109±.013 | 0.249±.000 | 0.247±.001 | 0.245±.001 |
| | TimeGAN | 0.132±.008 | 0.153±.014 | 0.220±.008 | 0.291±.003 | 0.303±.002 | 0.351±.004 |
| | TimeVAE | 0.118±.004 | 0.113±.005 | 0.110±.027 | 0.302±.001 | 0.318±.000 | 0.353±.003 |
| | Diffwave | 0.133±.008 | 0.129±.003 | 0.132±.001 | 0.252±.001 | 0.252±.000 | 0.251±.000 |
| | DiffTime | 0.118±.004 | 0.120±.008 | 0.118±.003 | 0.252±.000 | 0.251.±.000 | 0.251±.000 |
| | Cot-GAN | 0.135±.003 | 0.126±.001 | 0.129±.000 | 0.262±.002 | 0.269±.002 | 0.275±.004 |
| | Original | 0.114±.006 | 0.108±.005 | 0.106±.010 | 0.245±.002 | 0.243±.000 | 0.243±.000 |
| Context-FID Score↓ | SDformer-ar | **0.018±.003** | **0.024±.001** | **0.021±.001** | **0.031±.002** | **0.036±.002** | **0.041±.003** |
| | SDformer-m | 0.086±.008 | 0.094±.007 | 0.078±.006 | 0.160±.025 | 0.151±.011 | 0.136±.014 |
| | Diffusion-TS | 0.631±.058 | 0.787±.062 | 0.423±.038 | 0.135±.017 | 0.087±.019 | 0.126±.024 |
| | TimeGAN | 1.130±.102 | 1.553±.169 | 5.872±.208 | 1.230±.070 | 2.535±.372 | 5.032±.831 |
| | TimeVAE | 0.827±.146 | 1.062±.134 | 0.826±.093 | 2.662±.087 | 3.125±.106 | 3.768±.998 |
| | Diffwave | 1.543±.153 | 2.354±.170 | 2.899±.289 | 2.697±.418 | 5.552±.528 | 5.572±.584 |
| | DiffTime | 1.279±.083 | 2.554±.318 | 3.524±.830 | 0.762±.157 | 1.344±.131 | 4.735±.729 |
| | Cot-GAN | 3.008±.277 | 2.639±.427 | 4.075±.894 | 1.824±.144 | 1.822±.271 | 2.533±.467 |

Apart from unconditional generation, we also explore the performance of our proposed methods in conditional generation tasks. Our objective is to evaluate the model's versatility in handling both conditional and unconditional tasks. More specifically, we aim to train a single model that can effectively manage both unconditional and conditional tasks under different settings. Referring to [42], we set $\tau = 48$, $l = 8, 16, 24, 32$ in Equation (2), and then directly use the model trained under the unconditional generation task to complete the forecasting tasks under these different settings. Figure 3 displays several examples of forecasting tasks. The median values of forecasting are represented as the dotted line, and 5% and 95% quantiles are depicted as the shade areas (Green: SDformer-ar, Gray: Diffusion-TS). This demonstrates that SDformer-ar provides more reasonable forecasts with higher confidence compared to Diffusion-TS. Furthermore, more detailed results are illustrated in Figure 4. Based on these findings, the methods employing discrete token modeling demonstrates adaptability to both unconditional and conditional generation tasks of varying lengths without the need for retraining, while maintaining good performance.

## 5.4 Ablation study

To understand the contribution of each component to proposed methods, we conduct ablation experiments for two aspects, 1) The impact of vector quantization methods based on different measurements 2) The advantages of discrete representations in time series generation. More experimental results refer to Appendix C, due to limited space.

**Effect of similarity-driven vector quantization in Equation** (4)**.** For discrete token modeling method, the quality of discrete representations learning from continuous data determines the performance potential of the entire method, with vector quantization playing a crucial role. Therefore, we compare the impact of our proposed similarity-driven vector quantization with the commonly used distance-driven vector quantization on the overall method performance.

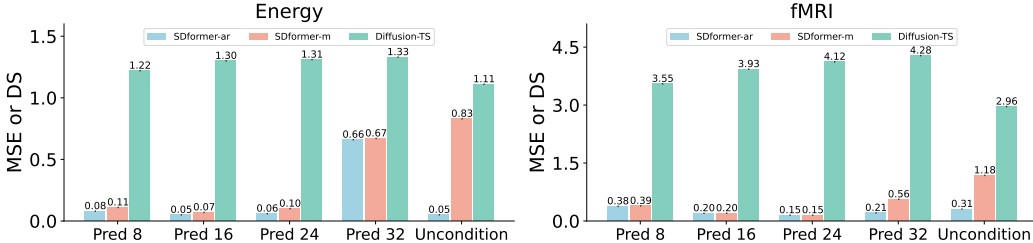

Figure 4: Performance for time series forecasting and generation under different setting. All forecasting tasks utilize Mean Square Error (MSE) as performance metric, while unconditional generation tasks employ Discriminative Scores (DS). Note: The data in this figure has been scaled by a factor of 100 for the forecasting tasks and 10 for the unconditional generation tasks to streamline the presentation.

Table 4: Results of ablation study.

| Metrics | Methods | Sines | Stocks | ETTh | MuJoCo | Energy | fMRI |
|---|---|---|---|---|---|---|---|
| Discriminative Score↓ | SDformer-ar | **0.006±.004** | **0.010±.006** | **0.003±.001** | **0.008±.005** | **0.006±.004** | **0.017±.007** |
| | w/o similarity | 0.006±.004 | 0.011±.007 | 0.010±.005 | 0.013±.003 | 0.018±.005 | 0.024±.003 |
| | continuous | 0.047±.012 | 0.065±.012 | 0.145±.020 | 0.055±.013 | 0.322±.012 | 0.243±.214 |
| | continuous, w/ first | 0.012±.004 | 0.021±.015 | 0.006±.004 | 0.020±.006 | 0.277±.007 | 0.074±.006 |
| | SDformer-m | **0.008±.004** | **0.020±.011** | **0.022±.001** | **0.025±.007** | **0.062±.006** | **0.043±.006** |
| | w/o similarity | 0.015±.007 | 0.081±.010 | 0.055±.004 | 0.070±.005 | 0.068±.005 | 0.055±.009 |
| Predictive Score↓ | SDformer-ar | **0.093±.000** | **0.037±.000** | **0.118±.002** | **0.007±.001** | **0.249±.000** | 0.091±.002 |
| | w/o similarity | **0.093±.000** | **0.037±.000** | 0.122±.002 | 0.008±.001 | **0.249±.000** | 0.091±.002 |
| | continuous | **0.093±.000** | 0.038±.000 | 0.124±.003 | 0.009±.001 | 0.255±.000 | 0.105±.000 |
| | continuous, w/ first | **0.093±.000** | **0.037±.000** | 0.122±.003 | **0.007±.001** | 0.251±.000 | **0.087±.003** |
| | SDformer-m | **0.093±.000** | **0.037±.000** | **0.119±.002** | **0.007±.001** | **0.250±.000** | **0.091±.001** |
| | w/o similarity | **0.093±.000** | **0.037±.000** | 0.123±.001 | 0.008±.001 | **0.250±.000** | 0.093±.000 |
| Context-FID Score↓ | SDformer-ar | **0.001±.000** | **0.002±.000** | 0.008±.001 | **0.005±.001** | **0.003±.000** | 0.015±.001 |
| | w/o similarity | 0.002±.000 | 0.012±.001 | 0.013±.001 | **0.005±.001** | 0.004±.000 | 0.011±.000 |
| | continuous | 0.056±.004 | 0.101±.022 | 0.433±.049 | 0.065±.008 | 0.213±.022 | 5.512±.390 |
| | continuous, w/ first | 0.004±.000 | 0.015±.003 | **0.002±.000** | 0.006±.001 | 0.021±.003 | **0.003±.000** |
| | SDformer-m | **0.010±.002** | **0.034±.008** | **0.019±.003** | **0.030±.003** | **0.041±.005** | **0.035±.003** |
| | w/o similarity | 0.044±.006 | 0.123±.009 | 0.106±.012 | 0.098±.009 | 0.062±.014 | 0.038±.002 |

The term "w/o similarity" in Table 4 denotes the variant of the corresponding method that employs distance-driven vector quantization instead of similarity-driven vector quantization. As per the comprehensive results, similarity-driven vector quantization significantly outperforms distance-driven vector quantization. For instance, the discriminative score witnessed an increase of 35.6% for SDformer-ar and 46.2% for SDformer-m.

**Effect of discrete token in Equation** (4) **and** (5). To explore the impact of discrete versus continuous tokens, we introduce a variant replacing the original discrete token with a continuous one within a similar framework. Initially, we substitute VQ-VAE with VAE as per [14] to encode a continuous latent space. In the second stage, we reconfigure the Transformer to accommodate continuous inputs, altering its initial input strategy due to the inapplicability of a fixed token like [BOS]. Thus, the first token of each time series is not used as the prediction target during training. For inference, we devised two methods: one involves sampling from a multivariate Gaussian, calculated from all initial training tokens, for generating the first token; the second uses the actual initial token directly. Although the latter does not lend itself to a fair comparison with the original model, it is included for a more comprehensive evaluation as a reference.

In Table 4, the term "continuous" represents the first variant, which involves using continuous tokens in place of discrete ones. Meanwhile, the term "continuous, w/ first" denotes the second variant, which builds upon the first by providing the actual first token during inference. It is worth noting that, as SDformer-m necessitates the utilization of category sampling during inference for achieving

iterative sampling, we abstain from conducting ablation experiments on discrete tokens specifically for this model. Based on the results, the model's performance experiences a significant decline upon the removal of discrete tokens. Even when incorporating the condition information of the first token, it often fails to surpass the original method in most scenarios.

# 6 Conclusions

In this paper, we present discrete token modeling for the time series generation tasks and propose a innovative two-stage model. Specifically, it is built upon an efficient time series tokenizer, which attains high-quality discrete token representations through similarity-driven vector quantization. Leveraging this foundation, we employ autoregressive token modeling and masked token modeling techniques to learn the distribution of time series data at the discrete token level. Experimental results showcase the efficacy and adaptability of our approach in various time series generation tasks. Owing to the flexibility of our method in the second stage, future work could explore referencing more efficient language models to design increasingly effective time series generation strategies.

## Acknowledgements

We would like to thank Tencent AI Lab for supporting Zhicheng Chen as a student researcher during his internship. The study was partially supported by the Key Laboratory of Data Protection and Intelligent Management, Ministry of Education, Sichuan University and also the Fundamental Research Funds for the Central Universities under Grant SCU2023D008.

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

# Supplementary Materials for SDformer

In the supplementary, we provide more implementation details, more experimental results, and visualization of test samples of our SDformer. We organize our supplementary as follows

- In Section A, we give the detailed description of used datasets and the metrics.

- In Section B, we provide the experiment settings.

- In Section C, we show more experimental results to verify the effectiveness and efficiency of SDformer.

- In Section D, we provide the details operations of the architecture in stage 1 and 2.

- In Section E, we provide the algorithms of training and inference in SDformer.

- In Section F, we specify the limitations of our method.

- In Section G, we showcase more visualization results and test samples on six TSG datasets.

## A  Dataset and metric details

**Dataset.**. The **Stocks** dataset consists of Google's stock price data between 2004 and 2019, with each observation representing a day and containing 6 features. The **ETTh** dataset includes data obtained from electrical transformers, encompassing load and oil temperature measurements taken every 15 minutes from July 2016 to July 2018. The **Energy** dataset, a UCI appliance energy prediction dataset, comprises 28 features. The **fMRI** dataset serves as a benchmark for causal discovery and features realistic simulations of blood-oxygen-level-dependent (BOLD) time series; we chose a simulation with 50 features from the original dataset referring to [42]. The **Sines** dataset contains 5 features, each generated independently with varying frequencies and phases. The **MuJoCo** dataset is a multivariate physics simulation time series dataset with 14 features.
**Metric**. The **Discriminative Score** quantifies the similarity between original and synthesized data. Initially, a classification model is trained using both the original and synthesized data. Subsequently, the model's capability to classify these data types is assessed, and the discriminative score is computed as $|Accuracy - 0.5|$. The **Predictive Score** evaluates the usefulness of the synthesized data by training a post-hoc sequence model to predict next-step temporal vectors using the train-synthesis-and-test-real (TSTR) method. The **Context-FID Score** quantifies the quality of the synthetic time series samples by computing the difference between representations of time series that fit into the local context.

## B  Experimental settings

Table 5: Detailed hyperparameters of SDformer.

| Dataset | Stage 1 | | | | | | Stage 2 | | | | |
|---|---|---|---|---|---|---|---|---|---|---|---|
| | Hidden dim | Enc/Dec Layers | $K$ | $d_c$ | $\lambda$ | r | Hidden dim | Transformer Layers | $\pi$ | N | S |
| Sines | 512 | 2 | 1024 | 512 | 0.5 | 4 | 1024 | 2 | 0.3 | 6 | [5,4,3,2,1,0] |
| Stocks | 512 | 2 | 512 | 256 | 2.0 | 4 | 1024 | 2 | 0.3 | 3 | [5,3,0] |
| ETTh | {512, 1024} | 2 | 512 | 512 | 0.5 | 4 | 1024 | 6 | 0.3 | 6 | [5,4,3,2,1,0] |
| MuJoCo | 512 | 2 | 512 | 512 | 0.5 | 4 | 1024 | {2,6} | 0.1 | 6 | [5,4,3,2,1,0] |
| Energy | 512 | 2 | 512 | 512 | 0.01 | 4 | 1024 | 2 | 0.1 | 6 | [5,4,3,2,1,0] |
| fMRI | 512 | {1,2} | 512 | 512 | 0.01 | {2,4} | 1024 | 2 | 0.1 | 6 | [5,4,3,2,1,0] |

In this part, we introduce our main experimental settings. For the unconditional generation task, we conduct five evaluations to obtain the experimental results. Similarly, for the conditional generation task, we run the process five times when computing the performance metrics, and escalate it to 1000 runs when calculating the median, along with the 5% and 95% quantiles of the predicted samples. Furthermore, we summarize the detailed hyperparameters of SDformer, shown as Table 5. The two values in {*,*} are the hyperparameters of SDformer-ar and SDformer-m respectively. Our primary experiments are executed on an Nvidia V-100 GPU with the AdamW [23] optimizer. In the future, we can use importance sampling [44] to further accelerate it.

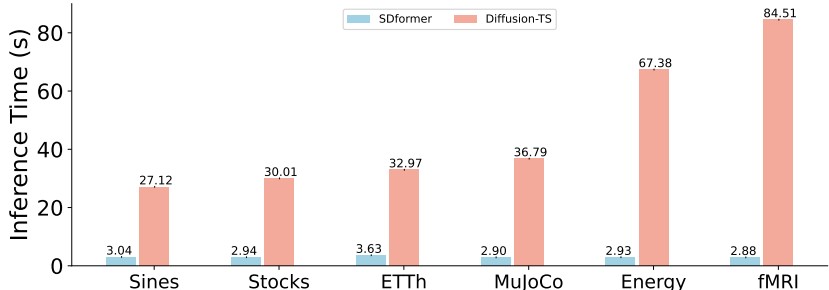

Figure 5: Inference time of SDformer and Diffusion-TS.

Table 6: Results of additional ablation study.

| Metrics | Methods | Sines | Stocks | ETTh | MuJoCo | Energy | fMRI |
|---|---|---|---|---|---|---|---|
| Discriminative Score↓ | SDformer-ar | **0.006±.004** | 0.010±.006 | **0.003±.001** | 0.008±.005 | **0.006±.004** | **0.017±.007** |
| | w/o Transformer | 0.007±.007 | **0.008±.005** | 0.007±.002 | **0.006±.003** | 0.009±.003 | 0.024±.009 |
| | SDformer-m | **0.008±.004** | **0.020±.011** | **0.022±.001** | **0.025±.007** | **0.062±.006** | **0.043±.006** |
| | w/o Transformer | 0.012±.007 | 0.033±.010 | 0.025±.007 | 0.027±.008 | 0.063±.009 | 0.060±.005 |
| Predictive Score↓ | SDformer-ar | **0.093±.000** | **0.037±.000** | **0.118±.002** | **0.007±.001** | **0.249±.000** | **0.091±.002** |
| | w/o Transformer | **0.093±.000** | **0.037±.000** | 0.119±.004 | 0.008±.001 | **0.249±.001** | 0.092±.003 |
| | SDformer-m | **0.093±.000** | **0.037±.000** | 0.119±.002 | **0.007±.001** | **0.250±.000** | **0.091±.001** |
| | w/o Transformer | **0.093±.000** | **0.037±.000** | **0.117±.003** | 0.008±.001 | **0.250±.000** | 0.093±.001 |
| Context-FID Score↓ | SDformer-ar | **0.001±.000** | **0.002±.000** | **0.008±.001** | **0.005±.001** | **0.003±.000** | **0.015±.001** |
| | w/o Transformer | 0.003±.000 | 0.003±.000 | 0.009±.001 | 0.006±.000 | 0.004±.000 | 0.018±.001 |
| | SDformer-m | **0.010±.002** | 0.034±.008 | **0.019±.003** | **0.030±.003** | 0.041±.005 | **0.035±.003** |
| | w/o Transformer | 0.013±.001 | **0.016±.002** | 0.045±.005 | 0.031±.005 | **0.038±.008** | 0.044±.002 |

# C Additional experimental results

**Comparison of inference time.** To validate the superiority of our method in terms of inference time over DDPMs-based approaches, we conduct a comparison between the inference times of SDformer and Diffusion-TS across all datasets, as illustrated in Figure 5. By integrating Figure 5 and Table 2, it can be demonstrated that our approach possesses a higher generation capability compared to the current DDPMs-based methods, while not necessitating the extensive inference time they require.

**Effect of Transformer architecture.** To investigate the contribution of the Transformer architecture to SDformer, we designed a variant that replaces the original Transformer architecture with an LSTM architecture. Specifically, for SDformer-ar, we employ a unidirectional LSTM [15] to replace the Decoder-only Transformer. Conversely, for SDformer-m, we utilize a bidirectional LSTM [30] to replace the Encoder-only Transformer, ensuring equivalent functionality. The term "w/o Transformer" in Table 6 denotes the variant of the corresponding method that employs LSTM architecture instead of Transformer architecture. Based on the results, the SDformer employing the LSTM architecture is somewhat less effective than the one based on the Transformer overall, but the difference is generally not substantial. We believe this is because the time series tokenizer, which is based on similarity-driven vector quantization in the first stage, has learned high-quality discrete token representations that are shorter and simpler than the original time series. As a result, even when using a simpler model in the second stage, it can still learn the data distribution of the time series quite effectively.

**Effect of model size.** To examine the impact of model size on SDformer, Table 7 illustrates the performance of SDformer across various model sizes. It is evident that with increasing model size, there is a noticeable enhancement in performance. Furthermore, to investigate whether our SDformer remains competitive with other methods when the model size is low, we reduce the model size by adjusting parameters such as the hidden dimension and code dimension, resulting in a smaller version of the SDformer (SDformer-s) that has a model size comparable to the current state-of-the-art baseline, Diffusion-TS. We then compare the performance of SDformer-s with Diffusion-TS, as demonstrated

Table 7: Performance discrepancy of SDformer across different model sizes on the Energy Dataset.

| Model Size (M) | 1.4 | 3.0 | 11.9 | 44.9 |
|---|---|---|---|---|
| Discriminative Score ↓ | 0.149±.007 | 0.084±.009 | 0.011±.009 | 0.006±.004 |
| Context-FID Score ↓ | 0.033±.003 | 0.022±.002 | 0.004±.000 | 0.003±.000 |

Table 8: The comparison results of small version SDformer (SDformer-s) against the baseline Diffusion-TS on Sines, Stocks and ETTh datasets.

| Metrics | Methods | Sines | Stocks | ETTh |
|---|---|---|---|---|
| Discriminative Score↓ | SDformer-s | **0.003±.003** | **0.019±.010** | **0.023±.001** |
| | Diffusion-TS | 0.006±.007 | 0.067±.015 | 0.061±.009 |
| Context-FID Score↓ | SDformer-s | **0.006±.000** | **0.015±.002** | **0.071±.001** |
| | Diffusion-TS | **0.006±.000** | 0.147±.025 | 0.116±.010 |
| Inference Time (s) | SDformer-s | 2.68 | 2.59 | 2.67 |
| | Diffusion-TS | 27.12 | 30.01 | 32.97 |
| Model Size (M) | SDformer-s | 0.17 | 0.16 | 0.28 |
| | Diffusion-TS | 0.24 | 0.29 | 0.35 |

in Table 8. We can observe that SDformer maintains competitive performance even with reduced model parameters and showcases notably faster inference times compared to Diffusion-TS.

# D   Model details

In this section, we present the detailed network architecture of the SDformer. The time series tokenizer in stage 1 primarily consists of 1D Convolution and 1D ResNet networks, as illustrated in Tables 9 and 10. In stage 2, the Autoregressive Transformer and Masked Transformer are implemented as standard decoder-only and encoder-only Transformer, respectively. The specifics of their blocks are depicted in Tables 11 and 12.

Table 9: The detailed architecture of the time serie tokenizer's encoder.

| Layer | Function | Descriptions |
|---|---|---|
| 1 | Convolution | input channel=$d$, output channel=D, kernel size=3, stride=1, padding=1 |
| 2 | ReLU | nn.ReLU() |
| 3 | Convolution | input channel=D, output channel=D, kernel size=4, stride=2, padding=1 |
| 4 | ResNet | input channel=D, depth=3, dilation growth rate=3 |
| 5 | ReLU | nn.ReLU() |
| 6 | Convolution | input channel=D, output channel=D, kernel size=4, stride=2, padding=1 |
| 7 | ResNet | input channel=D, depth=3, dilation growth rate=3 |
| 8 | ReLU | nn.ReLU() |
| 9 | Convolution | input channel= $d_c$, output channel=H, kernel size=3, stride=1, padding=1 |

# E   Algorithms

In this section, we detail the training and inference algorithms for SDformer. Specifically, Algorithm 1 outlines the training procedure for the time series tokenizer in the first stage, while Algorithm 2 and 3 elucidate the training processes for the Autoregressive Transformer and Masked Transformer in the second stage, respectively. Lastly, Algorithm 4 and 5 represent the inference procedures for SDformer-ar and SDformer-m, respectively.

Table 10: The detailed architecture of the time serie tokenizer's decoder.

| Layer | Function | Descriptions |
|---|---|---|
| 1 | Convolution | input channel=$d_c$, output channel=D, kernel size=3, stride=1, padding=1 |
| 2 | ReLU | nn.ReLU() |
| 3 | ResNet | input channel=D, depth=3, dilation growth rate=3 |
| 4 | ReLU | nn.ReLU() |
| 5 | Upsample | nn.Upsample() |
| 6 | Convolution | input channel=D, output channel=D, kernel size=3, stride=1, padding=1 |
| 7 | ResNet | input channel=D, depth=3, dilation growth rate=3 |
| 8 | ReLU | nn.ReLU() |
| 9 | Upsample | nn.Upsample() |
| 10 | Convolution | input channel=D, output channel=D, kernel size=3, stride=1, padding=1 |
| 11 | ReLU | nn.ReLU() |
| 12 | Convolution | input channel=D, output channel=D, kernel size=3, stride=1, padding=1 |
| 13 | ReLU | nn.ReLU() |
| 14 | Convolution | input channel=D, output channel=$d$, kernel size=3, stride=1, padding=1 |

Table 11: The detailed architecture of the Autoregressive Transformer block.

| Layer | Function | Descriptions |
|---|---|---|
| 1 | Layernorm | nn.LayerNorm() |
| 2 | Casual-attention | CasualAttention(q=x, k=x, v=x) |
| 3 | Layernorm | nn.LayerNorm() |
| 4 | MLP | nn.Linear() |
| 5 | ReLU | nn.ReLU() |
| 6 | MLP | nn.Linear() |

Table 12: The detailed architecture of the Masked Transformer block.

| Layer | Function | Descriptions |
|---|---|---|
| 1 | Layernorm | nn.LayerNorm() |
| 2 | Self-attention | Attention(q=x, k=x, v=x) |
| 3 | Layernorm | nn.LayerNorm() |
| 4 | MLP | nn.Linear() |
| 5 | ReLU | nn.ReLU() |
| 6 | MLP | nn.Linear() |

---

**Algorithm 1** Training of time series tokenizer.

---

**Input:** Time series dataset $D = \{X_{1:\tau}^i\}_{i=1}^n$
**Output:** Encoder $\mathcal{E}$, Decoder $\mathcal{D}$ and codebook $C$.

1: **for** $k \leftarrow 1$ to $K$ **do**
2:     Get the $X_{1:\tau} \sim D$;
3:     Feed the $X_{1:\tau}$ to Encoder $\mathcal{E}$ and get the $H_{1:L}$;
4:     Get the $Y_{1:L}$ by Equation (4);
5:     Get the $\tilde{H}_{1:L}$ by Equation (5);
6:     Feed the $\tilde{H}_{1:L}$ to Decoder $\mathcal{D}$ and get the $\tilde{X}_{1:\tau}$;
7:     Compute the training loss $\mathcal{L}$ by Equation (6);
8:     Complete backpropagation process based on $\mathcal{L}$ and update the parameters of $\mathcal{E}$ and $\mathcal{D}$;
9:     Use $H_{1:L}$ to update the codebook through exponential moving average;
10: **end for**
11: Return trained $\mathcal{E}$, $\mathcal{D}$, and $C$.

---

---

**Algorithm 2** Training of Autoregressive Transformer.

---

**Input:** Time series dataset $D = \{X_{1:\tau}^i\}_{i=1}^n$, optimized time series tokenizer.
**Output:** Autoregressive Transformer $\mathcal{G}_{ar}$.

---

1: **for** $k \leftarrow 1$ to $K$ **do**
2:     Get the discrete tokens $Y_{1:L}$ by $\mathcal{E}$ and Equation (4);
3:     Get the shifted tokens $Y_{1:L}^{in}$ by shifting $Y_{1:L}$ and appending [BOS] token;
4:     Update $Y_{1:L}^{in}$ by applying random replacement via Equation (8);
5:     Feed the $Y_{1:L}^{in}$ to Autoregressive Transformer $\mathcal{G}_{ar}$;
6:     Compute the training loss $\mathcal{L}_{ar}$ by Equation (7);
7:     Complete backpropagation process based on $\mathcal{L}_{ar}$ and update the parameters of $\mathcal{G}_{ar}$;
8: **end for**
9: Return trained $\mathcal{G}_{ar}$.

---

---

**Algorithm 3** Training of Masked Transformer.

---

**Input:** Time series dataset $D = \{X_{1:\tau}^i\}_{i=1}^n$, optimized time series tokenizer.
**Output:** Masked Transformer $\mathcal{G}_m$.

---

1: **for** $k \leftarrow 1$ to $K$ **do**
2:     Get the discrete tokens $Y_{1:L}$ by $\mathcal{E}$ and Equation (4);
3:     Get the masked tokens $\overline{Y}_{1:L}$ by randomly masking $Y_{1:L}$;
4:     Feed the $\overline{Y}_{1:L}$ to Masked Transformer $\mathcal{G}_m$;
5:     Compute the training loss $\mathcal{L}_{mask}$ by Equation (9);
6:     Complete backpropagation process based on $\mathcal{L}_{mask}$ and update the parameters of $\mathcal{G}_m$;
7: **end for**
8: Return trained $\mathcal{G}_m$.

---

# F   Limitations

If the goal is to achieve superior generation capabilities, the time series tokenizer must possess a relatively large codebook and a higher number of parameters. However, this will result in increased memory pressure.

# G   Additional visualizations

To visualize the performance of unconditional time series generation, we adopt three visualization methods: 1) projecting original and synthetic data in a 2-dimensional space using t-SNE [32]; 2) projecting original and synthetic data in a 2-dimensional space using Principal Component Analysis (PCA); 3) drawing data distributions using Kernel Density Estimation (KDE). Figure 3, along with Figures 6 through 8, display the visual outcomes of unconditional generation. It is evident that the

---

**Algorithm 4** Inference process of SDformer-ar.

---

**Require:** The token sequence length $L \in \mathbb{Z}$, codebook size $K \in \mathbb{Z}$.

1: $\overline{Y}^{(0)} \leftarrow [K]$;                                          $\triangleright$ [BOS] token as starting input.
2: **for** $t \in \{0, 1, \ldots, L-1\}$ **do**
3:     $L_{logits} \leftarrow \mathcal{G}_{ar}(\overline{Y}^{(t)})$;                       $\triangleright L_{logits} \in \mathbb{R}^{(t+1) \times K}$
4:     $p \leftarrow \text{softmax}(L_{logits}[-1])$;                    $\triangleright p \in \mathbb{R}^K$
5:     $I_{sampled} \leftarrow \text{Sample}(p)$;      $\triangleright$ Categorical sampling, $I_{sampled} \in \{0, \cdots, K-1\}$
6:     $\overline{Y}^{(t+1)} \leftarrow [\overline{Y}^{(t)}, I_{sampled}]$;        $\triangleright$ Concatenate the newly sampled token
7: **end for**
8: $\overline{H}_{1:L} = Q^{-1}(\overline{Y}^{(L)}[1:])$;     $\triangleright$ Dequantization in Equation (5), $\overline{H}_{1:L} \in \mathbb{R}^{L \times d_c}$
9: $\overline{X}_{1:\tau} = \mathcal{D}(\overline{H}_{1:L})$;                       $\triangleright \overline{X}_{1:\tau} \in \mathbb{R}^{\tau \times d}$
10: **return** $\overline{X}_{1:\tau}$.                               $\triangleright$ Return final output.

---

**Algorithm 5** Inference process of SDformer-m.

---

**Require:** The token sequence length $L \in \mathbb{Z}$, codebook size $K \in \mathbb{Z}$, iteration number $N \in \mathbb{Z}$, and mask schedule $S \in \mathbb{Z}^N$.

1: $\overline{Y}^{(0)} \leftarrow K \cdot \mathbf{1}_L$;           $\triangleright \mathbf{1}_L$ is an all-ones vector
2: **for** $t \in \{0, 1, \ldots, N-1\}$ **do**
3:      $L_{logits} \leftarrow \mathcal{G}_m(\overline{Y}^{(t)})$;        $\triangleright L_{logits} \in \mathbb{R}^{L \times K}$
4:      $p \leftarrow \text{softmax}(L_{logits})$;        $\triangleright p \in \mathbb{R}^{L \times K}$
5:      $I_{sampled} \leftarrow \text{Sample}(p)$;      $\triangleright$ Categorical sampling, $I_{sampled} \in \{0, \cdots, K-1\}^L$
6:      $p_s \leftarrow p[I_{sampled}]$;      $\triangleright p_s \in \mathbb{R}^L$ indicates the probability of the sampled tokens
7:      $p_s \leftarrow \text{where}(\overline{Y}^{(t)} \neq K, \mathbf{1}_L, p_s)$;      $\triangleright$ Assign a probability of 1 to unmasked tokens
8:      $T_{threshold} \leftarrow \text{sorted}(p_s)[S[t]+1]$;      $\triangleright$ Find the $(S[t]+1)$-th smallest value from $p_s$
9:      $I_{sampled} \leftarrow \text{where}(p_s < T_{threshold}, K \cdot \mathbf{1}_L, I_{sampled})$;      $\triangleright$ Mask low-probability tokens
10:     $\overline{Y}^{(t+1)} \leftarrow \text{where}(\overline{Y}^{(t)} \neq K, \overline{Y}^{(t)}, I_{sampled})$;      $\triangleright$ Update masked tokens
11: **end for**
12: $\overline{H}_{1:L} = Q^{-1}(\overline{Y}^{(N)})$;      $\triangleright$ Dequantization in Equation (5), $\overline{H}_{1:L} \in \mathbb{R}^{L \times d_c}$
13: $\overline{X}_{1:\tau} = \mathcal{D}(\overline{H}_{1:L})$;      $\triangleright \overline{X}_{1:\tau} \in \mathbb{R}^{\tau \times d}$
14: **return** $\overline{X}_{1:\tau}$.      $\triangleright$ Return final output

---

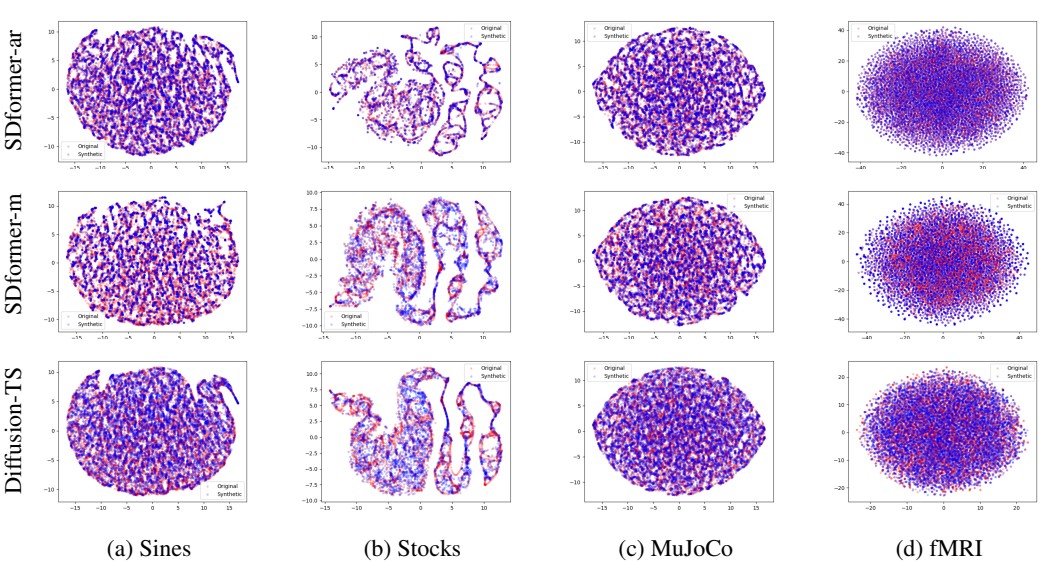

Figure 6: t-SNE visualizations of the time series synthesized by SDformer and Diffusion-TS on the Sines, Stocks, MuJoCo and fMRI datasets.

data generated by SDformer more closely aligns with the actual data compared to that produced by Diffusion-TS.

Figures 9 through 12 show the prediction samples of SDformer in different datasets and different prediction lengths. The median values of forecasting are represented as the dotted line, and 5% and 95% quantiles are depicted as the shade areas. As per the results, SDformer generates precise predictions in the majority of examples, signifying its strong performance in conditional generation tasks. Additionally, SDformer eliminates the need for retraining when undertaking these prediction tasks, emphasizing its remarkable versatility.

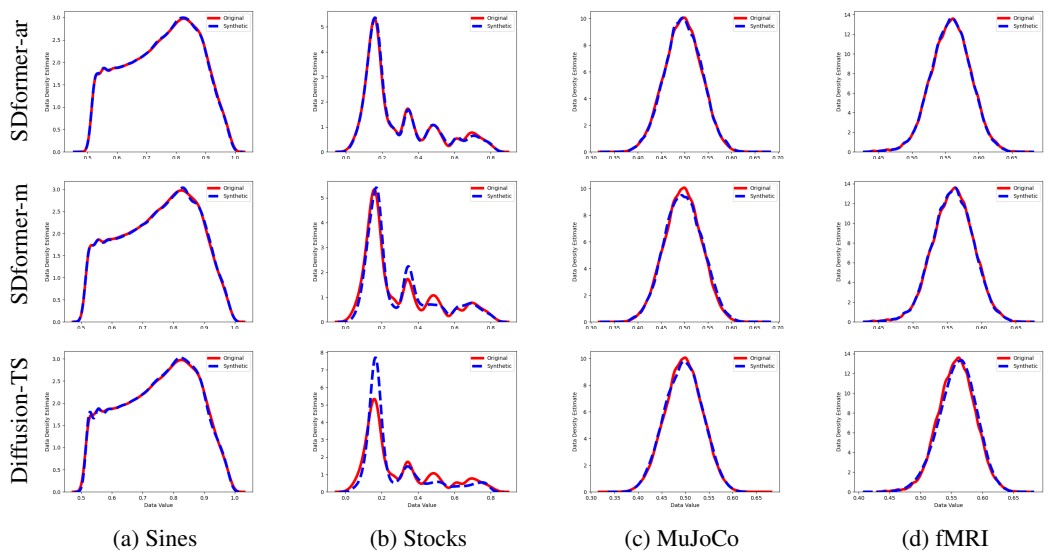

Figure 7: Kernel density estimation visualizations of the time series synthesized by SDformer and Diffusion-TS on the Sines, Stocks, MuJoCo and fMRI datasets..

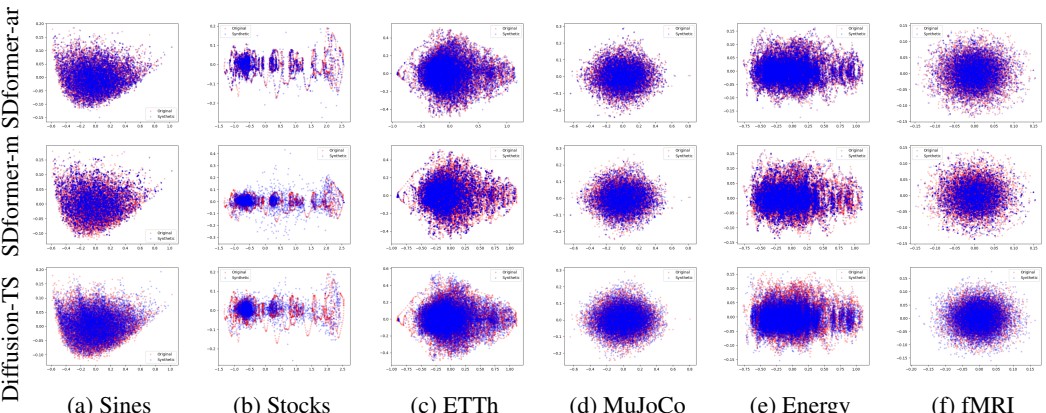

Figure 8: PCA visualizations of the time series synthesized by SDformer and Diffusion-TS across all datasets.

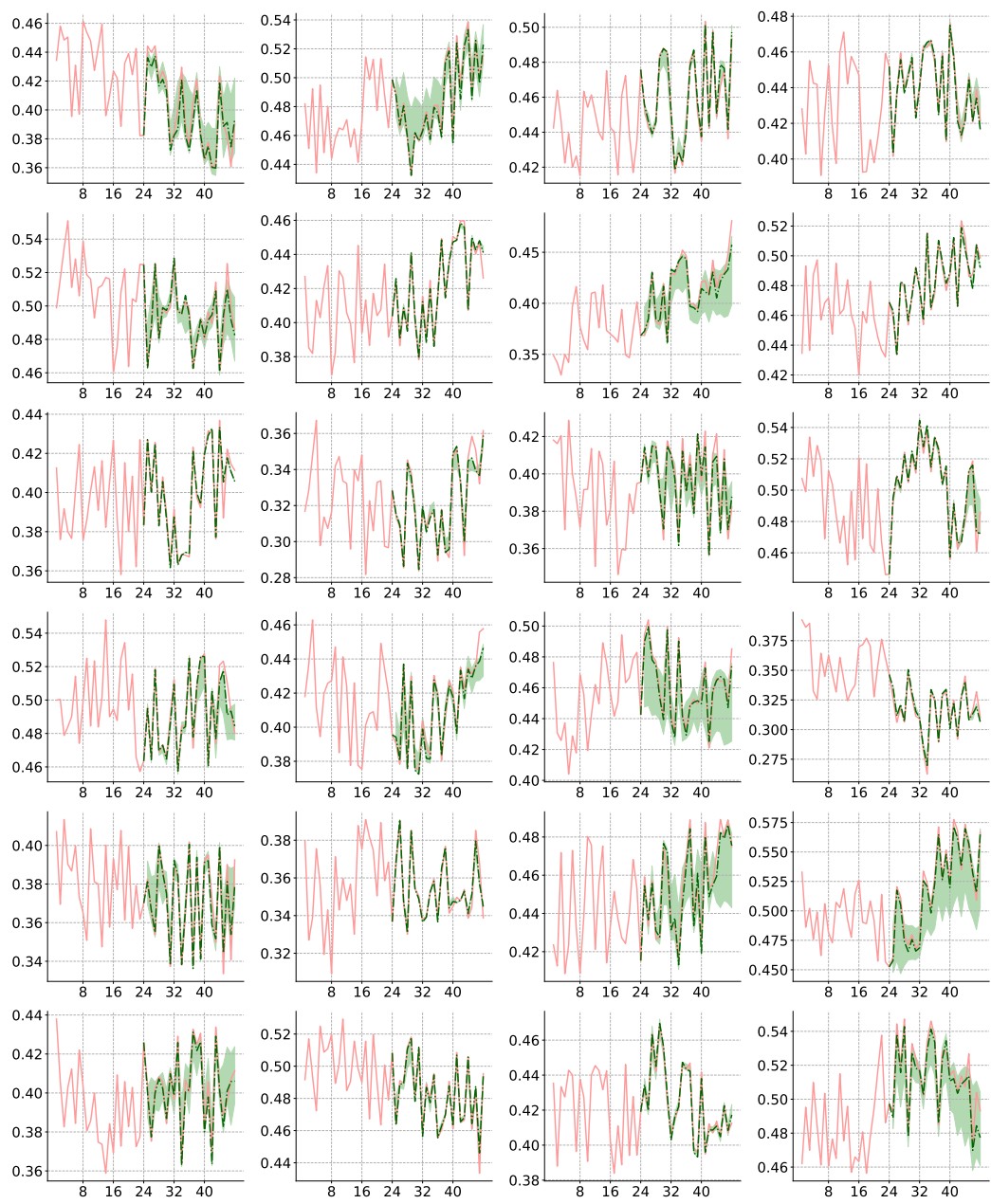

Figure 9: Examples of time series forecasting for the Energy dataset with a prediction length of 24. Green colors correspond to Predictions of SDformer.

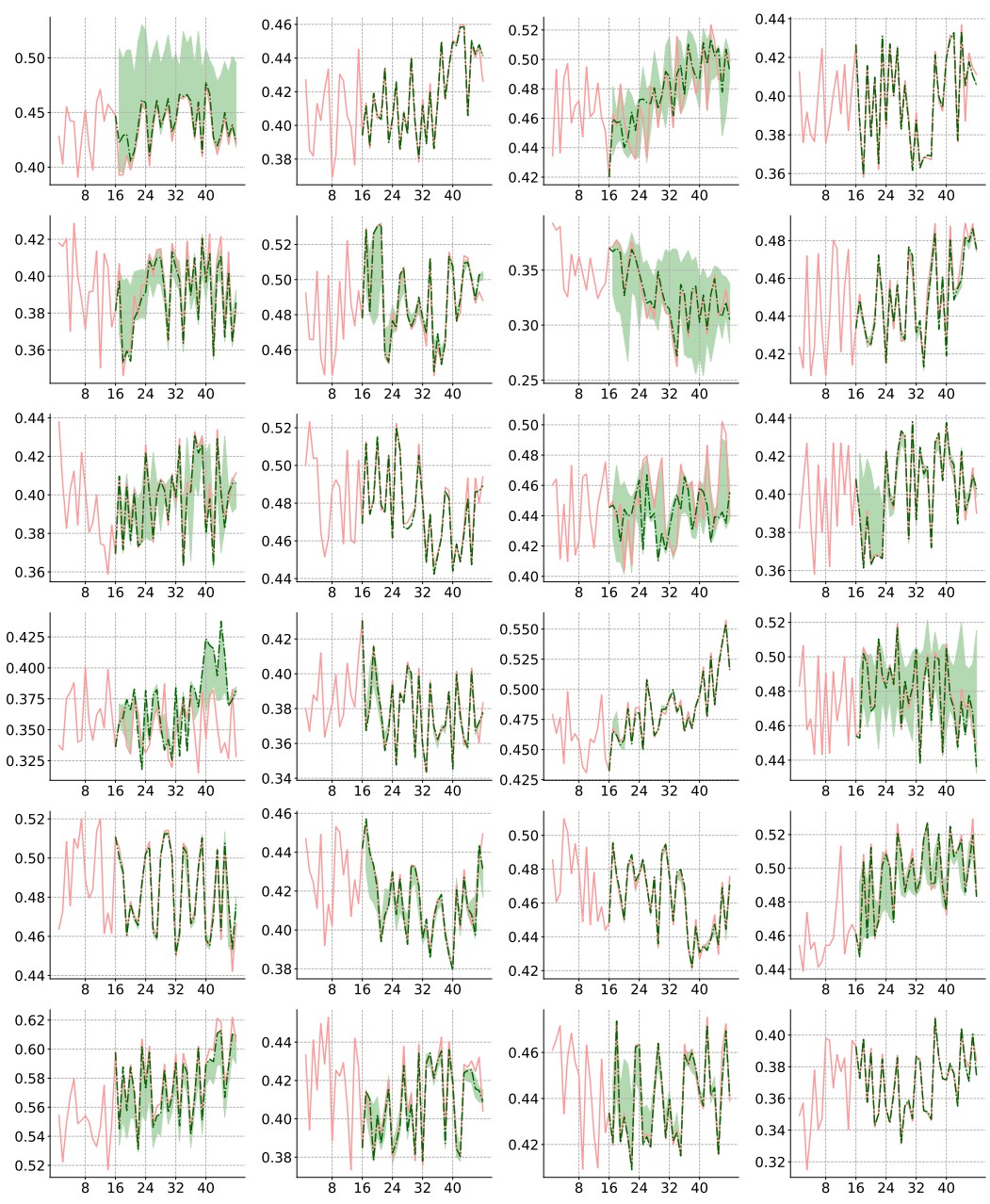

Figure 10: Examples of time series forecasting for the Energy dataset with a prediction length of 32. Green colors correspond to Predictions of SDformer.

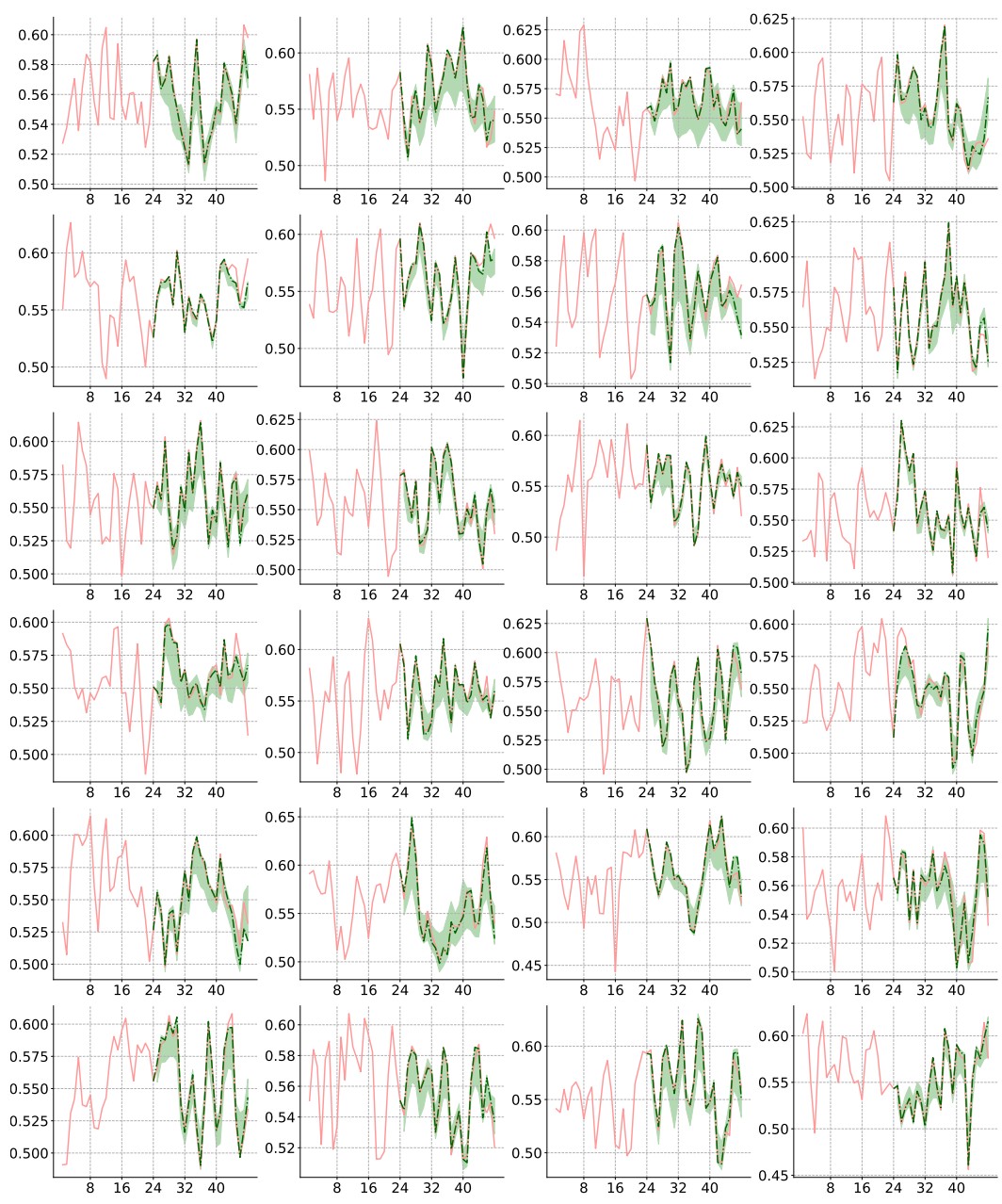

Figure 11: Examples of time series forecasting for the fMRI dataset with a prediction length of 24. Green colors correspond to Predictions of SDformer.

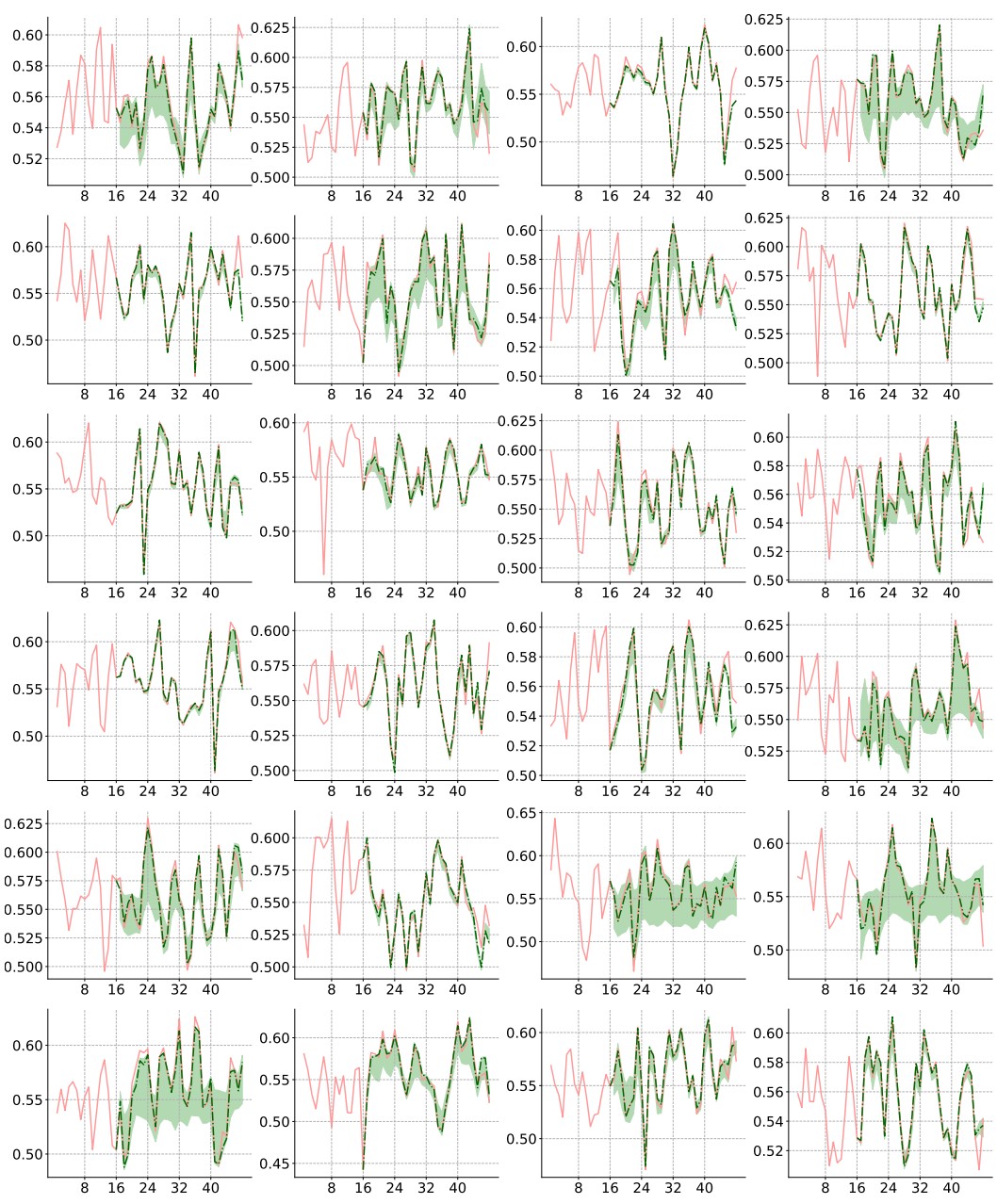

Figure 12: Examples of time series forecasting for the fMRI dataset with a prediction length of 32. Green colors correspond to Predictions of SDformer.

