# OpenReview forum: "SDformer: Similarity-driven Discrete Transformer For Time Series Generation"
_NeurIPS.cc/2024/Conference — NeurIPS 2024 poster_

### Official Review · Reviewer_DHCH · 2024-07-08

**Soundness:** 4
**Presentation:** 3
**Contribution:** 4
**Rating:** 7
**Confidence:** 4

**Summary:**

This paper investigates the use of a vector-quantization transformer for time series generation. The authors propose a similarity-driven vector quantization method for time series modeling. After training the encoder and decoder structures for generating embeddings, the authors propose two ways to generate time series: auto-regressive and masking. They empirically prove their idea across various settings, including different setups and datasets.

**Strengths:**

• The paper investigates the use of vector-quantization transformers, similar to MUSE or DALLE. This method is popular in LLMs; however, existing time series generation frameworks largely follow the image domain, such as diffusion models.

• The authors propose a similarity-driven vector quantization method, which is much more effective for time series than existing methods.

• The paper conducts experiments on various datasets, including well-known ones, demonstrating the effectiveness of the vector-quantization methods.

• The authors establish a time series forecasting task as a conditional time series generation and show the effectiveness of their methods.

**Weaknesses:**

Please refer to the Questions section.

**Questions:**

• First of all, the idea is novel and distinguishable from previous time series generation models. I wonder about the motivation behind this work.

• For conditional time series generation, not for imputation or forecasting where other time series values are conditioned, generating the time series for other conditions such as trend or global minimum. Can the authors suggest an idea to use their method for these kinds of generation tasks? In my opinion, VQ-based methods can be harder to control conditionally than diffusion frameworks. Please correct me if I have any misunderstandings.

• In terms of mask selection for masked token modeling, time series models often use special masking, such as geometric masking [1]. Randomly selected masks cannot fully learn the properties of the time series domain.

• In the proposed vector-quantized samples, is there any relationship between the distance of the embedding and the generated data samples, i.e., do small distances in vectors really exhibit similar time series data after decoding?

• Can the authors suggest more ablation study about model size and code embedding size?

[1] On the Constrained Time-Series Generation Problem, NeurIPS 2023

[2] A Transformer-based Framework for Multivariate Time Series Representation Learning, KDD 2021

**Limitations:**

The authors clarify the limitations in the Appendix.

---

> ### Author Rebuttal · Authors · 2024-08-07
>
> Thanks for your constructive reviews and suggestions. In the following, we will answer your questions one by one.
>
> **Q1.** I wonder about the motivation behind this work.
>
> **A1.** Our motivations include the following three points:
> * **Inference efficiency.** The inference time of previous SOTA models based on DDPMs is lengthy. To reduce this, we use vector quantization to compress the length of time series in latent space, significantly speeding up the second-stage Transformer. Additionally, non-autoregressive masked token modeling further decreases inference time.
> * **Generation quality.**  There is room to improve the quality of generated time series. To address this, we use similarity-driven vector quantization to capture correlations within time series, resulting in higher-quality discrete token representations. Additionally, we apply NLP techniques like autoregressive and masked token modeling to better learn time series distributions, effectively enhancing the quality of the generated time series.
> * **Compatibility with LLMs.** In recent years, LLM-related technologies have advanced rapidly. By aligning the format of time series data with natural language data (i.e., learning discrete time series tokens), we can directly leverage these advancements to tackle time series tasks. Our work confirms the feasibility of this approach and sets the stage for future research on integrating time series with LLM technology.
>
> **Q2.** Can the authors suggest an idea to use their method for other conditional time series generation, such as trend or global minimum?
>
> **A2.** You are correct that, compared to diffusion frameworks, VQ-based methods face greater challenges in handling conditional information, making this a valuable research direction. One approach could be to replace the [BOS] token in autoregressive token modeling with conditional input information from a conditional encoder. For instance, during training, we could input the trend or global minimum into the encoder and integrate it into the Transformer for joint training.
>
> **Q3.** In terms of mask selection for masked token modeling, time series models often use special masking, such as geometric masking. Randomly selected masks cannot fully learn the properties of the time series domain.
>
> **A3.** Thank you for your insight. Geometric masking is indeed more appropriate than random masking for capturing the properties of the time series domain. In time-domain datasets, a single missing point can often be accurately predicted using surrounding values, such as through interpolation, without requiring complex neural network modeling. Thus, such masking instances do not contribute to learning the time series data distribution. However, for SDformer-m, the processed data is a token sequence, where each token represents $\( r \times d \)$ data points from the original data space, with  $r$ being the temporal downsampling factor and $d$ the number of time variables. In this case, accurate prediction using adjacent tokens via interpolation or other methods is not feasible, necessitating neural network modeling. Additionally, we compare the impact of the two masking strategies on the Context-FID Score, as shown in the table below. The results indicate that random masking is more effective for token sequence modeling.
>
> | Method | randomly masking | geometric masking |
> |--------|------------------|-------------------|
> | Stocks | 0.034±.008       | 0.107±.006        |
> | Energy | 0.041±.005       | 0.065±.005        |
>
>
> **Q4.** In the proposed vector-quantized samples, is there any relationship between the distance of the embedding and the generated data samples, i.e., do small distances in vectors really exhibit similar time series data after decoding?
>
> **A4.** Yes, typically, the closer the latent vectors are, the more similar their decoded time series data will be. To validate this, we construct positive and negative sample pairs. Positive samples represent data pairs that belong to the same token after quantization (i.e., the latent vectors are similar), while negative samples represent data pairs that do not belong to the same token after quantization (i.e., the latent vectors are dissimilar). Subsequently, we compute the average time series distances of the positive and negative sample pairs after decoding, as shown in the table below. It is evident that the average distance of the positive sample pairs is significantly smaller than that of the negative sample pairs, indicating that the closer the latent vectors are, the more similar their decoded time series data will be.
>
> | Data             | Positive Sample Pairs | Negative Sample Pairs | Mixed Samples Pairs |
> |------------------|-----------------------|-----------------------|---------------------|
> | Average Distance | 0.005938              | 0.072874              | 0.072730            |
>
>
> **Q5.** Can the authors suggest more ablation study about model size and code embedding size?
>
> **A5.** Thank you for your suggestion. We will incorporate these ablation studies into the main text. The following two tables show the Context-FID Score of SDformer on the **Energy** dataset with varying model sizes and code embedding sizes, respectively. It can be observed that as the model size increases, the performance improves significantly. Moreover, a larger code embedding size also contributes to performance enhancement, albeit to a limited extent.
>
> | Model Size (M)     | 1.4        | 3.0        | 11.9       | 44.9       |
> |--------------------|------------|------------|------------|------------|
> | Context-FID Score↓ | 0.033±.003 | 0.022±.002 | 0.004±.000 | 0.003±.000 |
>
> | Code Embedding Size | 32           | 64           | 128          | 256          | 512          |
> |---------------------|--------------|--------------|--------------|--------------|--------------|
> | Context-FID Score↓  | 0.0039±.0001 | 0.0035±.0004 | 0.0029±.0004 | 0.0031±.0002 | 0.0026±.0002 |

---

> > ### Comment · Reviewer_DHCH · 2024-08-11
> > **Thanks for the response**
> >
> > I carefully read the authors' rebuttal, including the general response and answers to my questions and those of the other reviewers. After reading the rebuttal, I understood the authors' novel contributions and the experimental support provided. Furthermore, the method is distinct from existing time series generation models.
> >
> > Thus, I have raised the score.

---

> > > ### Author Response · Authors · 2024-08-11
> > > **Thanks to Reviewer DHCH**
> > >
> > > We sincerely appreciate the time and effort you spent on our work. Thank you!

---

### Official Review · Reviewer_2bQV · 2024-07-11

**Soundness:** 2
**Presentation:** 2
**Contribution:** 3
**Rating:** 6
**Confidence:** 3

**Summary:**

In this paper, the authors propose a time series generation model named SDformer. The model is based on the discrete token modeling techniques and demonstrate empirically its feasibility for the time-series generation task. The method presented in the paper surpasses the current state-of-the-art models on multiple metrics and datasets.

**Strengths:**

- In general, the paper presents interesting results that empirically show that the discretization of time series is very effective for the task of generation, which is a challenging problem.
- The method presents strong empirical results over various number of metrics.

**Weaknesses:**

Overall, employing transformers for time series data generation tasks is challenging, and there are only a few works in this direction.
There are some details that are missing and make this paper's results hard to reproduce (see Questions section).
My main concern is the lack of details in the paper and the missing of recent related work.

**Questions:**

- How did the authors choose the size of the codebook K?
- Code collapse is mentioned in the intro and not discussed further. What does it mean?
- Adding more precise descriptions for implementing the random replacement would be helpful for reproduction. For instance, how do you choose the value of $\pi$? Is it a hyperparameter?
- When zero-shot forecasting, it can be helpful to compare with some baseline. Is the diffusion-TS trained specifically in the conditional case? or is it compared to the same setting where you train for unconditional generation and perform zero-shot forecasting? The qualitative examples are nice but do not give us a thorough understanding of the model performance in that case.
- Why the unconditional results in Fig. 4 (bottom) have value > 0.5?
- For the long-range benchmark, I think it would be beneficial if the authors compare their results against [1]. The results of TimeVAE and TimeGAN can degrade as the length of the sequence grows merely due to the fact that they rely on LSTM\GRU at their backbone, which can be replaced easily for an encoder handling long-sequences.
- The authors do not compare the number of parameters for their original results. For a fair comparison, the authors should refer to Table 7 in the main text. In addition, can you add the results of the reduced size model on Energy and MuJoCo to Table 7 and compare the full model number parameters?
- Some time-series generation-related work and comparisons are missing. For instance, [2,3].
- I think the authors should also compare with [3]. Its performance is stronger than Diffusion-TS for some datasets and is much smaller regarding parameters and inference time.



[1] Zhou, Linqi, et al. "Deep latent state space models for time-series generation."

[2] Jeon, Jinsung, et al. "GT-GAN: General purpose time series synthesis with generative adversarial networks."

[3] Naiman, Ilan, et al. "Generative modeling of regular and irregular time series data via koopman VAEs."

**Limitations:**

See the Weaknesses and Questions sections.

---

> ### Author Rebuttal · Authors · 2024-08-07
>
> Thanks for your constructive reviews and suggestions. In the following, we will answer your questions one by one.
>
> **Q1.** How did the authors choose the size of the codebook K.
>
> **A1.** K is a hyperparameter that requires experimentation to determine the optimal value. Specifically, we choose from {128, 256, 512, 1024}. The table below presents the Discriminative Score of SDformer under different K. Our final choice can be found in Appendix B (Experimental Settings).
>
> | K      | 128        | 256         | 512         | 1024       |
> |--------|------------|-------------|-------------|------------|
> | Stocks  | 0.012±.004 | 0.010±.006  | 0.010±0.007 | 0.010±.007 |
> | Energy | 0.102±.012 | 0.017±.006  | 0.006±.004  | 0.005±.004 |
>
> **Q2.** Code collapse is mentioned in the intro and not discussed further.
>
> **A2.** Code collapse occurs when only a small part of the codebook gets updated during training, which degrades the performance of the VQ-VAE. The table below shows how code collapse happens on the Energy dataset due to the lack of EMA and Code Reset. I'll include it in the introduction.
>
> |                        | Codebook Usage (%) | Reconstruction loss (MSE) |
> |------------------------|--------------------|---------------------------|
> | w/ EMA and Code Reset  | 100                | 0.0001                    |
> | w/o EMA and Code Reset | 3.3                | 0.0061                    |
>
> **Q3.** Adding more precise descriptions for implementing the random replacement would be helpful for reproduction.
>
> **A3.** I'll revise the original text as follows: "In this approach, each token is processed individually. A random number is compared to a probability threshold, π. If it meets the threshold, the token is replaced randomly; otherwise, it remains unchanged." Pseudocode is provided below for clarification.
> ```plaintext
> if Uniform(0,I) <= π:
>     y_i <- randint(0,K)
> else:
>     y_i <- y_i
> ```
> In this pseudocode, Uniform(0,I) denotes a number randomly generated between 0 and 1, randint(0,K) signifies a token index randomly chosen from the codebook, and y_i represents the i-th token in the original sequence.
>
>
> **Q4.1.** When zero-shot forecasting, it can be helpful to compare with some baselines.
>
> **A4.1.** In our task setting, we focus on training for unconditional generation and performing zero-shot forecasting. As a result, it would be unjust to directly compare our approach with specialized forecasting models, which are typically trained exclusively on conditional generation tasks. However, if you have any suggested baselines, we would be more than happy to conduct a comparison with them.
>
> **Q4.2.** Is the diffusion-TS trained specifically in the conditional case? or is it compared to the same setting where you train for unconditional generation and perform zero-shot forecasting?
>
> **A4.2.** For the prediction task, our settings are the same, both training for unconditional generation and performing zero-shot forecasting.
>
> **Q4.3.** The qualitative examples are nice but do not give us a thorough understanding of the model performance in that case.
>
> **A4.3.** The qualitative examples in Fig. 3 and 9-12 are randomly selected instances from the prediction task, with their corresponding quantitative metrics shown in Fig. 4.
>
> **Q5.** Why the unconditional results in Fig. 4 (bottom) have value > 0.5?
>
> **A5.** Because the evaluation metrics for the prediction task and the unconditional generation task differ. To display them on the same bar chart, we have scaled the results accordingly, which is indicated in the caption of Fig. 4.
>
> **Q6.** For the long-range benchmark, comparing with LS4 would be beneficial.
>
> **A6.** Regarding LS4, we have attempted to reproduce the results but faced unforeseen environmental issues and time constraints. We are continuing to address these challenges and will update you promptly with any progress.
>
> **Q7.1.** The authors do not compare the number of parameters for their original results.
>
> **A7.1.** The parameter counts for our models are detailed in the following table, which we'll update in the main text.
>
> | Dataset        | Sines  | Stocks | ETTh   | MuJoCo | Energy | fMRI   |
> |----------------|--------|--------|--------|--------|--------|--------|
> | Model Size (M) | 45.9   | 44.4   | 95.3   | 44.9   | 44.9   | 36.9   |
>
>
> **Q7.2.** Can you add the results of the reduced size model on Energy and MuJoCo？
>
> **A7.2.** The table below presents the results of the reduced-size model on the Energy and MuJoCo datasets. It can be observed that the smaller SDformer model still maintains a performance advantage.
>
> |                      | Context-FID Score↓ | Model Size (M) |
> |----------------------|--------------------|----------------|
> | Energy: SDformer-s   | 0.033±.003         | 1.41           |
> | Energy: Diffusion-TS | 0.089±.003         | 1.14           |
> | MuJoCo: SDformer-s   | 0.008±.001         | 0.34           |
> | MuJoCo: Diffusion-TS | 0.013±.001         | 0.36           |
>
> **Q8.** Some time-series generation-related work and comparisons are missing, such as KoVAE and GT-GAN.
>
> **A8.** The table below shows the results of my reproduction of KoVAE and GT-GAN in the same environment. Some results differ from those in the original paper. We are in contact with the corresponding authors and will provide updates as they come.
>
> |               | Sines      | Stocks     | Energy     | MuJoCo     |
> |---------------|------------|------------|------------|------------|
> | DS: SDformer  | 0.006±.004 | 0.010±.006 | 0.006±.004 | 0.008±.005 |
> | DS: KoVAE     | 0.005±.004 | 0.027±.021 | 0.256±.025 | 0.052±.006 |
> | DS: GT-GAN    | 0.044±.025 | 0.012±.059 | 0.476±.006 | 0.246±.022 |
> | FID: SDformer | 0.001±.000 | 0.002±.006 | 0.003±.000 | 0.005±.005 |
> | FID: KoVAE    | 0.011±.001 | 0.074±.018 | 0.192±.035 | 0.034±.005 |
> | FID: GT-GAN   | 0.121±.029 | 0.171±.041 | 0.542±.062 | 0.547±.033 |
> (DS: Discriminative Score, FID: Context-FID Score)

---

> ### Comment · Reviewer_2bQV · 2024-08-12
>
> Thank you for your response and the additional results. Some of the results you reported for KoVAE and GT-GAN differ significantly from those in the original ICLR and NeurIPS papers. Additionally, the model parameter counts you provided (around 40M and sometimes even 100M) are much larger compared to competitive methods that have approximately 40K parameters.
> > ***and performing zero-shot forecasting. As a result, it would be unjust to directly compare our approach with specialized forecasting models, which are typically trained exclusively on conditional generation tasks.***
>
> If it does not perform well, what advantages does your model offer for forecasting?

---

> > ### Author Response · Authors · 2024-08-13
> > **Response to Reviewer 2bQV**
> >
> > **Q9.** Some of the results you reported for KoVAE and GT-GAN differ significantly from those in the original ICLR and NeurIPS papers.
> >
> > **A9.** Thank you for your feedback. Regarding KoVAE and GT-GAN, we have made our best efforts to optimize the hyperparameters and select the best-performing results. The inconsistency in some results may be due to more fine-grained hyperparameter choices, which are not available in the corresponding papers and codes. To present more accurate results, we are in contact with the corresponding authors and will continuously update the results.
> >
> > **Q10.** The model parameter counts you provided are much larger compared to competitive methods.
> >
> > **A10.** There might be some misunderstandings regarding the parameter counts in competitive methods. For instance, the parameter count of Diffusion-TS ranges from 200K to 1.4M. The following table investigates whether our SDformer remains competitive with other methods when operating with a reduced model size.
> >
> > |                              | Sines      | Stocks     | Etth       | MuJoCo     | Energy     |
> > |------------------------------|------------|------------|------------|------------|------------|
> > | SDformer-s: FID ↓            | 0.006±.000 | 0.015±.002 | 0.071±.001 | 0.008±.001 | 0.033±.003 |
> > | Diffusion-TS: FID ↓          | 0.006±.000 | 0.147±.025 | 0.116±.010 | 0.013±.001 | 0.089±.003 |
> > | SDformer-s: Model Size (M)   | 0.17       | 0.16       | 0.28       | 0.34       | 1.41       |
> > | Diffusion-TS: Model Size (M) | 0.24       | 0.29       | 0.35       | 0.36       | 1.14       |
> > (FID: Context-FID Score)
> >
> > The table below showcases the Context-FID Scores of SDformer and Diffusion-TS at varying parameter counts on the Energy dataset.
> >
> > | Model Size (M) | [1.1-1.5]  | [3.0-3.1]  | [11.9-12.1] | [44.9-48.0] |
> > |----------------|------------|------------|-------------|-------------|
> > | SDformer       | 0.033±.003 | 0.022±.002 | 0.004±.000  | 0.003±.000  |
> > | Diffusion-TS   | 0.089±.003 | 0.071±.023 | 0.060±.015  | 2.002±.339  |
> > (e.g. [1.1-1.5]:The model size is between 1.1M and 1.5M.)
> >
> > Overall, our method achieves significant performance regardless of whether it uses a small or large number of parameters. We appreciate your professional insights, we will investigate ways to maintain the model's accuracy while reducing the parameters in future work.
> >
> > **Q11.** If it does not perform well, what advantages does your model offer for forecasting?
> >
> > **A11.** In our **zero-shot forecasting** setup, following reference [1], we concentrate on training for unconditional generation tasks and conducting inference on prediction tasks, which means that we do not utilize any conditional information during the training process. The benefit of this approach is that we can flexibly adjust historical lengths and predictive lengths in the prediction task during the inference stage without the need to retrain the model.
> >
> > For **specialized forecasting models**, they often necessitate training various models for different historical lengths or predictive lengths, and subsequently performing inference under distinct length settings.
> >
> > Therefore, to ensure a fair comparison under the same settings, specifically training on unconditional generation tasks, we are limited to selecting models that support unconditional generation, such as Diffusion-TS 、Diffwave and DiffTime. Among the results presented, we select Diffusion-TS for comparison, since it demonstrates superior performance compared to Diffwave and DiffTime, as evidenced by reference [1]. In other words, it is not that our model underperforms compared to the specialized prediction model; rather, such a comparison is not suitable or relevant. Moreover, if you have any other suitable baselines in mind, we are open to including them in the comparison as well.
> >
> > [1] Xinyu Yuan and Yan Qiao. Diffusion-TS: Interpretable Diffusion for General Time Series Generation. The Twelfth International Conference on Learning Representations, {ICLR} 2024, Vienna, Austria, May 7-11, 2024.

---

### Official Review · Reviewer_t6Rv · 2024-07-12

**Soundness:** 3
**Presentation:** 3
**Contribution:** 3
**Rating:** 6
**Confidence:** 4

**Summary:**

This paper introduces SDformer, a method for time series generation, which addresses challenges in inference time and quality improvement. It utilizes a similarity-driven vector quantization technique to learn high-quality discrete token representations of time series. It employs a discrete Transformer for data distribution modeling at the token level. Experimental results demonstrate that SDformer outperforms competing approaches in generating high-quality time series while maintaining short inference times.

**Strengths:**

1. The idea of similarity-driven vector quantization approach is novel and makes sense.

2. The experiment conducted is adequate and the results are impressive. The analysis of the experimental results is detailed.

3. The paper is well-written and the presentation is good.

**Weaknesses:**

1. The description of the proposed method is not very detailed.

2. The limitations of the method have not been well explained. It should be compared with other methods, rather than just stating that achieving better generation results will lead to an increase in the number of parameters.

**Questions:**

As I stated above in the weakness, what's the exact limitation of the proposed model?

**Limitations:**

Yes.

---

> ### Author Rebuttal · Authors · 2024-08-07
>
> Thanks for your constructive reviews and suggestions. In the following, we will answer your questions one by one.
>
> **Q1.** The description of the proposed method is not very detailed.
>
> **A1.** Thank you for your feedback. SDformer is a two-stage method.
> * **First Stage:** In the first stage, we develop a variant of VQ-VAE [1], introducing an alignment method based on maximizing the similarity between discrete and continuous representations, abandoning the distance-minimization alignment approach used in VQ-VAE, as shown in Eq. 5. To further enhance the precision and stability of the training process, we apply Exponential Moving Average (EMA) for codebook updates and reset inactive codes (Code Reset) during training, which is included in line 48-50 of introduction. Also, the specific structures of the encoder and decoder in the first stage are presented in Tables 8 and 9 in Appendix D, and the detailed training process for the first stage is described in Algorithm 1.
> * **Second Stage:** In the second stage, we use the trained encoder and tokenizer in the first stage to obtain the discrete representation of the target time series, as shown on the right side of Figure 1. In this stage, we employ two different strategies: autoregressive (SDformer-ar) and non-autoregressive (SDformer-m) using a Transformer structure to accomplish the task of generating discrete indexes. The Transformer structures for the two different strategies in the second stage are presented in Tables 10 and 11 in Appendix D. The detailed processes for training and inference are described in Algorithms 2-5.
>
> To further address your concern, we would appreciate it if you could specify the aspects of the method description that you found unclear or insufficient. This will allow us to better understand your concerns and enhance the clarity and detail of our method description accordingly.
>
> **Q2.** The limitations of the method have not been well explained. It should be compared with other methods, rather than just stating that achieving better generation results will lead to an increase in the number of parameters.
>
> **A2.** Thank you for your suggestion. The table below shows the Context-FID Score of SDformer and Diffusion-TS at varying parameter quantities. While SDformer demonstrates competitive performance with smaller parameters, it’s evident that there is significant potential for improvement as the parameters increase. This suggests that achieving higher performance may require managing increased memory demands. Conversely, using smaller parameters could result in a notable drop in performance.
>
> On the other hand, Diffusion-TS shows a gradual performance improvement initially, but with a large enough parameter quantity, its performance may become distorted. This indicates that one can opt for smaller parameters in Diffusion-TS without a substantial loss in performance. We will incorporate these findings into our paper to better address our limitations. Your insights are highly appreciated and play a crucial role in improving our manuscript.
>
> | Model Size (M) | [1.1-1.5]  | [3.0-3.1]  | [11.9-12.1] | [44.9-48.0] |
> |----------------|------------|------------|-------------|-------------|
> | SDformer       | 0.033±.003 | 0.022±.002 | 0.004±.000  | 0.003±.000  |
> | Diffusion-TS   | 0.089±.003 | 0.071±.023 | 0.060±.015  | 2.002±.339  |
> (e.g. [1.1-1.5]:The model size is between 1.1M and 1.5M.)
>
> **Q3.** What's the exact limitation of the proposed model?
>
> **A3.** The SDformer has a slightly larger parameters to ensure high accuracy, which results in relatively higher memory requirements. In future work, we will explore how to maintain the model's accuracy while reducing the parameters.
>
> [1]. Van Den Oord, Aaron, and Oriol Vinyals. "Neural discrete representation learning." Advances in neural information processing systems 30 (2017).

---

### Official Review · Reviewer_h9mP · 2024-07-12

**Soundness:** 2
**Presentation:** 2
**Contribution:** 2
**Rating:** 5
**Confidence:** 2

**Summary:**

This paper proposed a MAE type attention model for time series generation. One major advantage is its efficiency, which is much faster than the diffusion based method. Various experiments are conducted and the proposed model outperform all baselines.

**Strengths:**

1. Efficiency: The paper introduces SDformer, a novel approach to time series generation that addresses key limitations of Denoised Diffusion Probabilistic Models (DDPMs). The proposed model demonstrates superior computational efficiency compared to diffusion-type models, which is a significant advancement in the field.

2. Methodology: SDformer's two-step process, combining similarity-driven vector quantization and discrete Transformer modeling, presents an innovative solution to the challenges of long inference times and quality improvement in time series generation.

3. Empirical Results: The numerical results presented in the paper are compelling, showing significant improvements over competing approaches in terms of generated time series quality and inference time. This empirical evidence strongly supports the effectiveness of the proposed method.

**Weaknesses:**

1. Reproducibility: One concern of the current submission is the lack of sample code provided to reviewers for verifying the reported experimental results.

**Questions:**

Please see the weaknesses part.

**Limitations:**

Please see the weaknesses part.

At the current stage, this paper proposed an interesting model in time series generation and I tend to recommend acceptance. However, I am open to reconsidering this assessment based on the further discussions.

---

> ### Author Rebuttal · Authors · 2024-08-07
>
> Thanks for your constructive reviews and suggestions. In the following, we will answer your question.
>
> **Q1.** One concern of the current submission is the lack of sample code provided to reviewers for verifying the reported experimental results.
>
> **A1.** Thank you for your suggestion. We have provided the anonymous link containing the sample code to the AC. This will allow you to directly verify our experimental results.

---

> > ### Author Response · Authors · 2024-08-08
> > **Code link  for SDformer reproduction**
> >
> > Thank you for your work. We have obtained AC's permission to directly publish anonymous code link containing some experimental  examples. If you have any questions, you can tell us directly.
> >
> > Code link: https://anonymous.4open.science/r/SDformer-main/

---

### Decision · Program_Chairs · 2024-09-25

**Decision:**

Accept (poster)

**Comment:**

Strengths:

- The proposed method, SDformer, addresses key limitations of existing DDPM-based models for time series generation - it improves inference time and generation quality.
- SDformer utilizes a novel similarity-driven vector quantization approach to learn high-quality discrete token representations of time series, followed by a discrete Transformer for data distribution modeling. This is a novel and promising approach.
- The paper is generally well-written and the presentation is good.

Weaknesses:
- The description of the proposed method is not very detailed, and some aspects could be explained better for reproducibility.
- The limitations of the method, such as the relationship between model size/parameters and performance, have not been fully explored.
- There are some discrepancies between the results reported for baseline methods (KoVAE, GT-GAN) and the original papers.
- The comparison to specialized time series forecasting models is limited, as the paper focuses on unconditional generation.

Based on the overall positive reviews and the novel contributions of the work, I would recommend accepting this paper. The authors have provided satisfactory responses to the reviewers' questions and concerns, and have agreed to incorporate more details and analysis into the final version.